# ABSTRACTING AND REFINING PROVABLY SUFFICIENT EXPLANATIONS OF NEURAL NETWORK PREDICTIONS

## ABSTRACT

Despite significant advancements in post-hoc explainability techniques for neural networks, many current methods rely on approximations and heuristics and do not provide formally provable guarantees over the explanations provided. Recent work has shown that it is possible to obtain explanations with formal guarantees by identifying subsets of input features that are sufficient to determine that predictions remain unchanged by incorporating neural network verification techniques. Despite the appeal of these explanations, their computation faces significant scalability challenges. In this work, we address this gap by proposing a novel abstraction-refinement technique for efficiently computing provably sufficient explanations of neural network predictions. Our method *abstracts* the original large neural network by constructing a substantially reduced network, where a sufficient explanation of the reduced network is also *provably sufficient* for the original network, hence significantly speeding up the verification process. If the explanation is insufficient on the reduced network, we iteratively *refine* the network size (by gradually increasing it) until convergence. Our experimental results demonstrate that our approach substantially enhances the efficiency of obtaining provably sufficient explanations for neural network predictions while additionally providing a fine-grained interpretation of the network's decisions across different abstraction levels. We thus regard this work as a substantial step forward in improving the feasibility of computing explanations with formal guarantees for neural networks.

## 1 INTRODUCTION

Despite the widespread use of deep neural networks, they remain black boxes that are uninterpretable to humans. Various methods have been proposed to explain neural network predictions. Classic additive feature attributions like LIME (Ribeiro et al., 2016), SHAP (Lundberg & Lee, 2017), and integrated gradients (Sundararajan et al., 2017) assume that neural networks exhibit near-linear behavior in a local region around the interpreted instance. Following these works, methods like Anchors (Ribeiro et al., 2018) and SIS (Carter et al., 2019) aim to compute a subset of input features that is (nearly) sufficient to determine the prediction. We refer to this subset of features as an *explanation* of the prediction. A common assumption in the literature is that a *smaller* explanation provides a better interpretation, and for this reason, the minimality of the explanation is also a desired property (Ignatiev et al., 2019; Carter et al., 2019; Darwiche & Hirth, 2020; Ribeiro et al., 2018; Barceló et al., 2020).

Methods like Anchors and SIS rely on probabilistic sampling of the input space and lack formally provable guarantees for the sufficiency of the subsets they identify. In contrast, recent approaches have demonstrated that incorporating neural network verification tools can produce explanations that are provably certified as sufficient (Bassan & Katz, 2023; Wu et al., 2024; La Malfa et al., 2021). This makes such explanations more suitable for safety-critical domains where formally certifying the reliability of the explanation is vital (Marques-Silva & Ignatiev, 2022).

Despite the appeal of such explanations, the computational complexity of producing them limits their feasibility for large neural networks (Barceló et al., 2020). Verifying a single query on a neural network is NP-Complete (Katz et al., 2017), with exponentially increasing complexity based on the number of nonlinear activations. Consequently, larger and deeper neural networks are significantly harder to verify. While there have been rapid advances in the scalability of neural

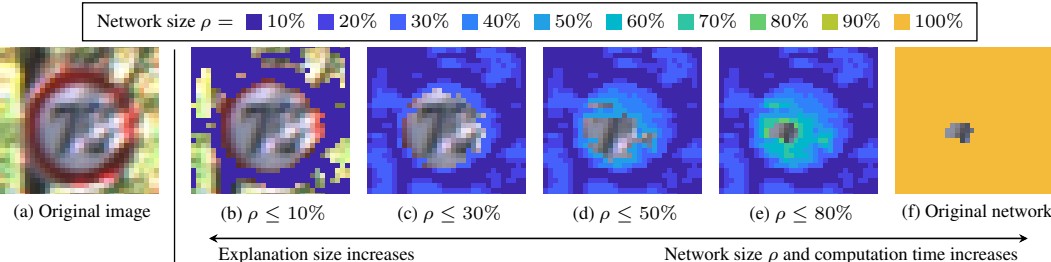

Figure 1: Demonstration of an abstraction-based explanation process. As the size of the abstract network $\rho$ increases, the size of the explanation (uncolored pixels) decreases. Notably, the majority of the explanation can be derived using only a small percentage of the network (b)-(e), reducing the time required to compute the explanation and offering more insight compared to using only the original network (f). Further visualizations are provided in appendix C.

network verification techniques in recent years (Wang et al., 2021; Brix et al., 2023), scalability remains a major challenge. Furthermore, providing *minimal* sufficient explanations (with respect to their sizes) requires invoking not one but multiple verification queries: for example, methods suggested by previous research (Bassan & Katz, 2023; Wu et al., 2024) dispatch a linear number of such queries relative to the input dimension, making these tasks particularly difficult for large input spaces.

**Our Contributions.** In this work, we propose a novel algorithm that significantly enhances the efficiency of generating provably sufficient explanations for neural networks. Our algorithm is based on an *abstraction refinement* technique (Clarke et al., 2000; Wang et al., 2006; Flanagan & Qadeer, 2002), which is widely used to improve the scalability of formal verification and model checking. In abstraction-refinement, a complex model with many states is efficiently optimized through two steps: (i) *abstraction*, which simplifies the model by grouping states, and (ii) *refinement*, which increases the precision of the abstraction to better approximate the model.

In the context of explainability, we propose an algorithm that constructs an *abstract neural network* — a substantially reduced model compared to the original. This reduction is achieved by merging neurons within each layer that exhibit similar behavior. The key component of this approach is to design the reduction such that a sufficient explanation for the abstract network is also provably sufficient for the original network. Hence, we define an explanation for the abstract neural network as an *abstract explanation*. Verifying the sufficiency of explanations for an abstract neural network is much more efficient than for the original model due to its reduced dimensionality. However, if a subset is found to be insufficient for the abstract network, its sufficiency for the original model is undetermined. Consequently, while sufficient explanations for the abstract network will be sufficient for the original network, *minimal* sufficient explanations for the abstract network, though sufficient, may not be minimal for the original network.

To address this issue, we incorporate a refinement component as typical in abstraction-refinenemt techniques: We construct an intermediate abstract network, which is slightly larger than the initial abstract network but still significantly smaller than the original. The explanations computed for this refined network are still provably sufficient for the original network and are also guaranteed to be a subset of the explanation from the initial abstract network. Hence, this phase produces a *refined explanation* based on the refined network. After several refinement steps, the sizes of the neural networks will gradually increase while the sizes of the explanations will gradually decrease until finally converging to a minimal explanation for some reduced network, which is also provably minimal for the original neural network. An illustration of this entire process is shown in Fig. 1.

We evaluate our algorithm on various benchmarks and show that it significantly outperforms existing verification-based methods by producing much smaller explanations and doing so substantially more efficiently. Additionally, we compare our results to heuristic-based approaches and show that these methods do not provide sufficient explanations in most cases, whereas the explanations of our approach are guaranteed to be sufficient. An additional advantage of our method is that it

enables the progressive convergence of the refined explanations to the final explanation, as illustrated in Fig. 1. This approach allows practitioners to observe minimal subsets across various reduced networks, offering a fine-grained interpretation of the model's prediction. Additionally, it provides the possibility of halting the process once the explanation meets some desired criteria.

Besides these practical aspects, we view this work as a novel proof-of-concept for using abstraction-refinement-based techniques in explainability, by obtaining formally provable explanations over abstract neural networks, which allow significantly more efficient verification, and a fine-grained interpretation over abstracted and refined networks. We hence consider this work as a significant step in the exploration of producing explanations with formal guarantees for neural networks.

## 2 PRELIMINARIES

### 2.1 NEURAL NETWORK VERIFICATION

We specify a generic neural network classifier architecture that can utilize any element-wise nonlinear activation function. For an input $\boldsymbol{x} \in \mathbb{R}^n$, the neural network classifier is denoted as $f \colon \mathbb{R}^n \to \mathbb{R}^c$. Numerous tools have been proposed for formally verifying properties of neural networks, with adversarial robustness being the most frequently examined property (Brix et al., 2023). The neural network verification query can be formalized as follows:

---

**Neural Network Verification (Problem Statement)**:
**Input**: A neural network $f$, such that $f(\boldsymbol{x}) = \boldsymbol{y}$, with an input specification $\psi_{\text{in}}(\boldsymbol{x})$, and and unsafe output specification $\psi_{\text{out}}(\boldsymbol{y})$.
**Output**: *No*, if there exists some $\boldsymbol{x} \in \mathbb{R}^n$ such that $\psi_{\text{in}}(\boldsymbol{x})$ holds and for $\boldsymbol{y} = f(\boldsymbol{x})$: $\psi_{\text{out}}(\boldsymbol{y})$ holds, and *Yes* otherwise.

---

There exist many off-the-shelf neural network verifiers (Brix et al., 2023). If the input specifications $\psi_{in}(\boldsymbol{x})$, output specifications $\psi_{out}(\boldsymbol{y})$, and model $f$ are piecewise-linear (e.g., $f$ uses ReLU activations), this task can be solved exactly (Katz et al., 2017). For non-piecewise-linear activations like sigmoid, the output is usually enclosed by bounding all approximation errors (Singh et al., 2018).

### 2.2 FORMALLY PROVABLE MINIMAL SUFFICIENT EXPLANATIONS

In this study, we concentrate on local post-hoc explanations for neural network classifiers. Specifically, for a neural network classifier $f \colon \mathbb{R}^n \to \mathbb{R}^d$ and a given local input $\boldsymbol{x} \in \mathbb{R}^n$ that has been assigned to class $t := \arg\max_j f(\boldsymbol{x})_j$, our objective is to explain why $f(\boldsymbol{x})$ was classified as class $t$.

**Sufficient Explanations.** A common method for interpreting the decisions of classifiers involves identifying subsets of input features $\mathcal{S} \subseteq [n]$ such that fixing these features to their specific values guarantees the prediction remains constant. Specifically, these techniques guarantee that the classification result remains consistent across *any* potential assignment within the complementary set $\bar{\mathcal{S}}$, thereby allowing for the formal validation of the explanations' soundness. While in the classic setting approaches, the complementary set $\bar{\mathcal{S}} := [n] \setminus \mathcal{S}$ is allowed to take on any possible feature values (Ignatiev et al., 2019; Darwiche & Hirth, 2020; Bassan & Katz, 2023), a more feasible and generalizable version restricts the possible assignments for $\bar{\mathcal{S}}$ to a bounded $\epsilon_p$-region (Wu et al., 2024; La Malfa et al., 2021; Izza et al., 2024). We use $(\boldsymbol{x}_{\mathcal{S}}; \tilde{\boldsymbol{x}}_{\bar{\mathcal{S}}}) \in \mathbb{R}^n$ to denote an assignment where the features of $\mathcal{S}$ are set to the values of the vector $\boldsymbol{x} \in \mathbb{R}^n$ and the features of $\bar{\mathcal{S}}$ are set to the values of another vector $\tilde{\boldsymbol{x}} \in \mathbb{R}^n$. Formally, we define a sufficient explanation $\mathcal{S}$ as follows:

**Definition 1 (Sufficient Explanation)** *Given a neural network $f$, an input $\boldsymbol{x} \in \mathbb{R}^n$, a perturbation radius $\epsilon_p \in \mathbb{R}$, and a subset $\mathcal{S} \subseteq [n]$, we say that $\mathcal{S}$ is a sufficient explanation concerning the query $\langle f, \boldsymbol{x}, \mathcal{S}, \epsilon_p \rangle$ on an $\ell_p$-norm ball $B_p^{\epsilon_p}$ of radius $\epsilon_p$ around $\boldsymbol{x}$ iff it holds that:*

$$\forall \tilde{\boldsymbol{x}} \in B_p^{\epsilon_p}(\boldsymbol{x}) \colon \quad [\arg\max_j f(\boldsymbol{x}_{\mathcal{S}}; \tilde{\boldsymbol{x}}_{\bar{\mathcal{S}}})_{(j)} = \arg\max_j f(\boldsymbol{x})_{(j)}],$$

$$with \quad B_p^{\epsilon_p}(\boldsymbol{x}) := \{\tilde{\boldsymbol{x}} \in \mathbb{R}^n \mid \|\boldsymbol{x} - \tilde{\boldsymbol{x}}\|_p \leq \epsilon_p\}.$$

*We define:* $\text{suff}(f, \boldsymbol{x}, \mathcal{S}, \epsilon_p) = 1$ *iff $\mathcal{S}$ constitutes a sufficient explanation with respect to the query $\langle f, \boldsymbol{x}, \mathcal{S}, \epsilon_p \rangle$; otherwise,* $\text{suff}(f, \boldsymbol{x}, \mathcal{S}, \epsilon_p) = 0$.

Def. 1 can be formulated as a neural network verification query. This method has been proposed by prior studies, which employed these techniques to validate the sufficiency of specific subsets (Wu et al., 2024; Bassan & Katz, 2023; La Malfa et al., 2021).

**Minimal Explanations.** Clearly, if the subset $\mathcal{S}$ is chosen as the entire input set, i.e., $\mathcal{S} := [n]$, it is a sufficient explanation. However, a common view in the literature suggests that smaller subsets are more meaningful than larger ones (Ribeiro et al., 2018; Carter et al., 2019; Barceló et al., 2020; Ignatiev et al., 2019). Therefore, there is a focus on identifying subsets that not only are sufficient but also meet a criteria for minimality:

**Definition 2 (Minimal Sufficient Explanation)** *Given a neural network $f$, an input $\boldsymbol{x} \in \mathbb{R}^n$, and a subset $\mathcal{S} \subseteq [n]$, we say that $\mathcal{S}$ is a minimal sufficient explanation concerning the query $\langle f, \boldsymbol{x}, \mathcal{S}, \epsilon_p \rangle$ on an $\ell_p$-norm ball $B_p^{\epsilon_p}$ of radius $\epsilon_p$ iff $\mathcal{S}$ is a sufficient explanation, and for any $i \in [n]$, $\mathcal{S} \setminus \{i\}$ is not a sufficient explanation. We define $\mathrm{min\text{-}suff}(f, \boldsymbol{x}, \mathcal{S}, \epsilon_p) = 1$ if $\mathcal{S}$ satisfies both sufficiency and minimality concerning $\langle f, \boldsymbol{x}, \mathcal{S}, \epsilon_p \rangle$, and $\mathrm{min\text{-}suff}(f, \boldsymbol{x}, \mathcal{S}, \epsilon_p) = 0$ otherwise.*

Minimal sufficient explanations can also be determined using neural network verifiers. Unlike simply verifying the sufficiency of a specific subset, this process requires executing multiple verification queries to ensure the minimality of the subset. Alg. 1 outlines such a procedure (similar methods are discussed in (Ignatiev et al., 2019; Wu et al., 2024; Bassan & Katz, 2023)). The algorithm begins with $\mathcal{S}$ encompassing the entire feature set $[n]$ and iteratively tries to exclude a feature $i$ from $\mathcal{S}$, each time checking whether $\mathcal{S} \setminus \{i\}$ remains sufficient. If $\mathcal{S} \setminus \{i\}$ is still sufficient, feature $i$ is removed; otherwise, it is retained in the explanation. This process is repeated until a minimal subset is obtained.

---

**Algorithm 1** Greedy Minimal Sufficient Explanation Search

---

**Input:** Neural network $f \colon \mathbb{R}^n \to \mathbb{R}^c$, input $\boldsymbol{x} \in \mathbb{R}^n$, perturbation radius $\epsilon_p \in \mathbb{R}$

1: $\mathcal{S} \leftarrow [n]$           $\triangleright$ Current sufficient explanation
2: **for each** feature $i \in [n]$ by some ordering **do**    $\triangleright$ The invariant $\mathrm{suff}(f, \boldsymbol{x}, \mathcal{S}, \epsilon_p)$ holds
3:    **if** $\mathrm{suff}(f, \boldsymbol{x}, \mathcal{S} \setminus \{i\}, \epsilon_p)$ **then**    $\triangleright$ Validated by a neural network verifier
4:      $\mathcal{S} \leftarrow \mathcal{S} \setminus \{i\}$
5:    **end if**
6: **end for**
7: **return** $\mathcal{S}$           $\triangleright$ $\mathrm{min\text{-}suff}(f, \boldsymbol{x}, \mathcal{S}, \epsilon_p)$ holds

---

## 3 FROM ABSTRACT NEURAL NETWORKS TO ABSTRACT EXPLANATIONS

A primary challenge in obtaining minimal sufficient explanations in neural networks is the high computational complexity involved. Verifying the sufficiency of a subset through a neural network verification query is NP-Complete (Katz et al., 2017), with complexity increasing exponentially with the number of activations, making it especially difficult for larger networks (Brix et al., 2023). Obtaining a minimal subset also requires a linear number of verification queries relative to the input size (Ignatiev et al., 2019; Wu et al., 2024), making the process computationally intensive for large inputs. One potential solution is to replace the original neural network $f$ with a much smaller, *abstract* neural network $f'$, and then run verifying queries on $f'$ instead of $f$. However, a key challenge here is to ensure that a sufficient explanation for $f'$ is also a sufficient explanation for $f$. Although abstraction techniques have been applied to improve the efficiency of adversarial robustness verification (Elboher et al., 2020; Liu et al., 2024; Ladner & Althoff, 2023), to our knowledge, we are the first to use such an approach to obtain provable explanations for neural networks.

**Abstract Neural Networks.** When abstracting a neural network, rather than using a traditional network $f \colon \mathbb{R}^n \to \mathbb{R}^c$, it is common to employ an abstract neural network $f'$ that outputs a set that encloses the actual output of $f$ (Ladner & Althoff, 2023; Prabhakar & Rahimi Afzal, 2019; Boudardara et al., 2022). This approach facilitates a more flexible propagation of the network's inner bounds capturing the error due to the abstraction. More formally, we define the domain of our abstract network $f' \colon \mathbb{R}^n \to 2^{(\mathbb{R}^c)}$, where $2^{(\mathbb{R}^c)}$ denotes the power set of $\mathbb{R}^c$. In the simplest case, this means that our abstract network outputs a $c$-dimensional interval rather than a $c$-dimensional vector.

Since our abstract network $f'$ now outputs a set, we must define a sufficient explanation for an abstract network. Specifically, we define a sufficient explanation for $f'$ and a target class $t \in [c]$ as a subset $\mathcal{S} \subseteq [n]$ such that when the features in $\mathcal{S}$ are fixed to their values in $\boldsymbol{x}$, the lower bound for the target class $t$ consistently exceeds the upper bound of all other classes $j \neq t$:

**Definition 3 (Sufficient Explanation for Abstract Network)** $\mathcal{S} \subseteq [n]$ *is a sufficient explanation of an abstract network $f'$ concerning the query $\langle f', \boldsymbol{x}, \mathcal{S}, \epsilon_p \rangle$ iff*

$$\forall j \neq t \in [c], \forall \tilde{\boldsymbol{x}} \in B_p^{\epsilon_p}(\boldsymbol{x}): \quad [\min(f'(\boldsymbol{x}_{\mathcal{S}}; \tilde{\boldsymbol{x}}_{\bar{\mathcal{S}}})_{(t)}) \geq \max(f'(\boldsymbol{x}_{\mathcal{S}}; \tilde{\boldsymbol{x}}_{\bar{\mathcal{S}}})_{(j)})],$$

$$\text{with} \quad B_p^{\epsilon_p}(\boldsymbol{x}) := \{\tilde{\boldsymbol{x}} \in \mathbb{R}^n \mid \|\boldsymbol{x} - \tilde{\boldsymbol{x}}\|_p \leq \epsilon_p\}.$$

**Neuron-Merging-Based Abstraction.** Various strategies can be employed to abstract a neural network to reduce its size. In this work, we use a fully-automatic abstraction technique (Ladner & Althoff, 2023), which involves merging neurons that exhibit similar behavior within the neural network for some bounded input set. For instance, numerous sigmoid neurons may become fully saturated, producing outputs close to $1$. Hence, the corresponding abstraction approach involves merging these saturated neurons and establishing corresponding error bounds for the given input set. This can be realized without large computational overhead to a desired reduction rate $\rho \in [0, 1]$ such that the overall verification time, including abstraction, mainly depends on the remaining number of neurons. For convenience, we also say that setting $\rho = 1$ produces the original network. We give details about the construction of the abstract network in Appendix A.

We are now prepared to establish the following claim concerning sufficient explanations concerning the query $\langle f', \boldsymbol{x}, \mathcal{S}, \epsilon_p \rangle$:

**Corollary 1 (Explanation Under Abstraction)** *Given a neural network $f$, an input $\boldsymbol{x}$, a perturbation radius $\epsilon_p$, a subset $\mathcal{S} \subseteq [n]$, let $f'$ be an abstract network constructed by neuron merging concerning the query $\langle f, \boldsymbol{x}, \mathcal{S}, \epsilon_p \rangle$. Then, it holds that:*

$$\text{suff}(f', \boldsymbol{x}, \mathcal{S}, \epsilon_p) \implies \text{suff}(f, \boldsymbol{x}, \mathcal{S}, \epsilon_p).$$

*Proof.* The proof can be found in Appendix A. $\square$

However, while a sufficient explanation $\mathcal{S}$ for the query $\langle f', \boldsymbol{x}, \mathcal{S}, \epsilon_p \rangle$ is also provably sufficient for the query $\langle f, \boldsymbol{x}, \mathcal{S}, \epsilon_p \rangle$, if $\mathcal{S}$ is insufficient for $\langle f', \boldsymbol{x}, \mathcal{S}, \epsilon_p \rangle$, it does not necessarily mean it is insufficient for $\langle f, \boldsymbol{x}, \mathcal{S}, \epsilon_p \rangle$. To more clearly highlight the explanation $\mathcal{S}$ within the context of the abstract network $f'$, we introduce an intermediate type of explanation, termed an *abstract sufficient explanation*. This is a provably sufficient explanation for $\langle f', \boldsymbol{x}, \mathcal{S}, \epsilon_p \rangle$ and, by extension (Cor. 1), also provably sufficient for $\langle f, \boldsymbol{x}, \mathcal{S}, \epsilon_p \rangle$:

**Definition 4 (Abstract Sufficient Explanation)** *We define a sufficient explanation $\mathcal{S}$ concerning the query $\langle f', \boldsymbol{x}, \mathcal{S}, \epsilon_p \rangle$ as an abstract sufficient explanation concerning the query $\langle f, \boldsymbol{x}, \mathcal{S}, \epsilon_p \rangle$.*

However, despite the fact that we now have a framework to obtain sufficient explanations $\mathcal{S}$ for a neural network $f$ much more efficiently as any query on the smaller abstract network is faster, there still remains a problem that the explanation $\mathcal{S}$ produced over the abstract network $f'$ may not be minimal over the original network $f$ even if it is minimal with respect to $\langle f', \boldsymbol{x}, \mathcal{S}, \epsilon_p \rangle$. We show how to address this issue through a refinement process for both the neural network and the explanation, which is carried out iteratively until convergence.

## 4 FROM REFINING NEURAL NETWORKS TO REFINING EXPLANATIONS

In order to produce an explanation that is both sufficient and minimal, we apply an iterative refinement process. In each step, we construct a slightly larger refined network $f''$ than the previously constructed abstract network $f'$ by splitting some of the merged neurons, resulting in a larger reduction rate $\rho'' > \rho'$. The refined abstract network $f''$ is still substantially smaller than the original network $f$ but slightly larger than $f'$, allowing us to generate a smaller explanation:

**Proposition 1 (Refined Abstract Network)** *Given a neural network $f$, an input $\boldsymbol{x}$, a perturbation radius $\epsilon_p \in \mathbb{R}$, a subset $\mathcal{S} \subseteq [n]$, and an abstract network $f'$ with reduction rate $\rho' \in [0, 1]$, we can construct a refined abstract network $f''$ from $f, f'$ with reduction rate $\rho'' > \rho'$, for which holds that:*

$$\forall \tilde{\boldsymbol{x}} \in B_p^{\epsilon_p}(\boldsymbol{x}): f(\boldsymbol{x}_{\mathcal{S}}; \tilde{\boldsymbol{x}}_{\bar{\mathcal{S}}}) \in f''(\boldsymbol{x}_{\mathcal{S}}; \tilde{\boldsymbol{x}}_{\bar{\mathcal{S}}}) \subset f'(\boldsymbol{x}_{\mathcal{S}}; \tilde{\boldsymbol{x}}_{\bar{\mathcal{S}}}).$$

*Proof.* The proof can be found in Appendix A. $\qquad\square$

Considering a refined abstract network $f''$ in relation to $f$ and $f'$ with $\rho'' > \rho'$, the following property holds for the explanations generated for these networks:

**Corollary 2 (Intermediate Sufficient Explanation)** *Let there be a neural network $f$, an abstract network $f'$, and a refined neural network $f''$. Then, it holds that:*

$$\text{suff}(f', \boldsymbol{x}, \mathcal{S}, \epsilon_p) \implies \text{suff}(f'', \boldsymbol{x}, \mathcal{S}, \epsilon_p) \text{ and}$$
$$\text{suff}(f'', \boldsymbol{x}, \mathcal{S}, \epsilon_p) \implies \text{suff}(f, \boldsymbol{x}, \mathcal{S}, \epsilon_p).$$

*Proof.* The proof can be found in Appendix A. $\qquad\square$

We observe that any sufficient explanation $\mathcal{S}$ for the query $\langle f'', \boldsymbol{x}, \mathcal{S}, \epsilon_p \rangle$ is also sufficient for the query $\langle f, \boldsymbol{x}, \mathcal{S}, \epsilon_p \rangle$ and might not be for $\langle f', \boldsymbol{x}, \mathcal{S}, \epsilon_p \rangle$ (Cor. 2). Thus, the intermediate explanation of a refined network is a subset of the explanation of the abstract network. Consequently, we define, in a manner akin to abstract sufficient explanations, an intermediate category termed *refined sufficient explanations*:

**Definition 5 (Refined Sufficient Explanation)** *We define a sufficient explanation $\mathcal{S}$ concerning $\langle f'', \boldsymbol{x}, \mathcal{S}, \epsilon_p \rangle$ as a refined sufficient explanation, where the refined abstract network $f''$ is constructed with respect to $\langle f, f', \boldsymbol{x}, \mathcal{S}, \epsilon_p \rangle$ for a neural network $f$, and an abstract network $f'$.*

The observation that a sufficient explanation for $\langle f'', \boldsymbol{x}, \mathcal{S}, \epsilon_p \rangle$ is a subset of the one for $\langle f', \boldsymbol{x}, \mathcal{S}, \epsilon_p \rangle$ suggests the following characteristic about the minimality of sufficient explanations:

**Corollary 3 (Intermediate Minimal Sufficient Explanation)** *Let there be a neural network $f$, an abstract network $f'$, and a refined neural network $f''$. Then, if $\mathcal{S}$ is a sufficient explanation concerning $f$, $f'$, and $f''$, it holds that:*

$$\text{min-suff}(f, \boldsymbol{x}, \mathcal{S}, \epsilon_p) \implies \text{min-suff}(f'', \boldsymbol{x}, \mathcal{S}, \epsilon_p) \text{ and}$$
$$\text{min-suff}(f'', \boldsymbol{x}, \mathcal{S}, \epsilon_p) \implies \text{min-suff}(f', \boldsymbol{x}, \mathcal{S}, \epsilon_p).$$

*Proof.* The proof can be found in Appendix A. $\qquad\square$

We note that the implication chain in Cor. 3 is in reverse order compared to the implication chain in Cor. 2. Intuitively, refining the abstract network $f'$ to $f''$ incrementally produces larger neural networks, which in turn generates progressively smaller explanations, until ultimately converging to a minimal explanation for some refined network as it converges to the original network. We harness this iterative process and propose an abstraction-refinement approach to produce such minimal subsets. The pseudo-code is given in Alg. 2.

The algorithm starts with a coarse abstract network $f'$ and derives an abstract sufficient explanation $\mathcal{S}$ by progressively removing features from $\mathcal{S}$, akin to the method described in Alg. 1. All following line numbers are with respect to Alg. 2. While the abstract sufficient explanation is provably sufficient for the original network (Cor. 2), it is not necessarily provably minimal (Cor. 3). If we cannot be certain whether a subset $\mathcal{S}$ is sufficient for the abstract network (lines 9 to 18), we check if feature $i$ is indeed not in the explanation by extracting a counterexample and checking its output of the original network (lines 10 to 11). If the counterexample is spurious due to the abstraction in $f'$, we refine the abstract neural network and thus produce a slightly larger network $f''$ (line 15). Using this refined network $f''$, we acquire a refined sufficient explanation relative to it, which allows us to remove more features from the explanation as the abstraction error is smaller (Prop. 1). As we remove more features from the explanation, the verification query to test whether the current subset is a sufficient explanation becomes harder as more features can be perturbed. Thus, it is sensible to only abstract the network in line 4 to a level for which the verification query was still successful, i.e., use the reduction rate of $f''$. This process continues, with each iteration slightly enlarging the abstract network through refinement and consequently reducing the size of the sufficient explanation.

**Proposition 2 (Greedy Minimal Sufficient Explanation Search)** *Alg. 2 produces a provably sufficient and minimal explanation $\mathcal{S}$ concerning the query $\langle f, \boldsymbol{x}, \mathcal{S}, \epsilon_p \rangle$, which converges to the same explanation as obtained by Alg. 1.*

*Proof.* The proof can be found in Appendix A. $\qquad\square$

---

**Algorithm 2** Greedy Minimal Abstract Sufficient Explanation Search

---

**Input:** Neural network $f : \mathbb{R}^n \to \mathbb{R}^c$, input $\boldsymbol{x} \in \mathbb{R}^n$, perturbation radius $\epsilon_p \in \mathbb{R}$

1: $\mathcal{S} \leftarrow [n]$               $\triangleright$ Current sufficient explanation
2: $\mathcal{F} \leftarrow [n]$               $\triangleright$ Features left to iterate on
3: **for each** feature $i \in \mathcal{F}$ by some ordering **do**    $\triangleright$ The invariant suff$(f, \boldsymbol{x}, \mathcal{S}, \epsilon_p)$ holds
4:    Abstract $f$ w.r.t suff$(f, \boldsymbol{x}, \mathcal{S} \setminus \{i\}, \epsilon_p)$ to get $f'$
5:    **do**
6:      **if** suff$(f', \boldsymbol{x}, \mathcal{S} \setminus \{i\}, \epsilon_p)$ **then**          $\triangleright$ Def. 3
7:        $\mathcal{S} \leftarrow \mathcal{S} \setminus \{i\}$       $\triangleright$ Feature $i$ is not in the explanation
8:        **break**
9:      **else**
10:        Extract counterexample $\tilde{\boldsymbol{x}}$ w.r.t suff$(f', \boldsymbol{x}, \mathcal{S} \setminus \{i\}, \epsilon_p)$
11:        **if** $\arg\max_j f(\tilde{\boldsymbol{x}})_{(j)} \neq \arg\max_j f(\boldsymbol{x})_{(j)}$ **then**      $\triangleright$ Def. 1
12:          $\mathcal{F} \leftarrow \mathcal{F} \setminus \{i\}$      $\triangleright$ Feature $i$ is in the explanation
13:          **break**
14:        **else**              $\triangleright$ Abstraction too coarse
15:          $f'' \leftarrow$ Refine $f'$ w.r.t suff$(f', \boldsymbol{x}, \mathcal{S} \setminus \{i\}, \epsilon_p)$    $\triangleright$ Prop. 1
16:          $f' \leftarrow f''$     $\triangleright$ Use this reduction rate in future interations
17:        **end if**
18:      **end if**
19:    **while** true            $\triangleright$ Repeat with refined network
20: **end for**
21: **return** $\mathcal{S}$            $\triangleright$ min-suff$(f, \boldsymbol{x}, \mathcal{S}, \epsilon_p)$ holds

---

## 5   EXPERIMENTAL RESULTS

**Experimental Setup.** We implemented the algorithms using CORA (Althoff, 2015) as the backend neural network verifier. We performed our experiments on three image classification benchmarks: (a) MNIST (LeCun, 1998), (b) CIFAR-10 (Krizhevsky et al., 2009), and (c) GTSRB (Stallkamp et al., 2012). Comprehensive details about the models and their training are provided in Appendix C.

### 5.1   COMPARISON TO STANDARD VERIFICATION-BASED EXPLANATIONS

In our initial experiment, we aimed to evaluate the abstraction-refinement method proposed in Alg. 2 against the traditional approach described in Alg. 1 for deriving provably sufficient explanations for neural networks, as implemented in previous studies (see (Wu et al., 2024; Bassan & Katz, 2023; La Malfa et al., 2021)). Complete details about the implementation of the refinement process are available in Appendix B. We assessed the effectiveness of both approaches using the two most prevalent metrics for evaluating sufficient explanations, as documented in (Wu et al., 2024; Ignatiev et al., 2019; Bassan & Katz, 2023): (i) the size of the explanation, with smaller sizes indicating higher meaningfulness, and (ii) the computation time.

Fig. 2 illustrates the reduction in explanation size over time for each of the three benchmarks. We observed a notable improvement in computation time using the abstraction-refinement approach compared to the standard greedy search method ($-41\%$ for MNIST, $-36\%$ for CIFAR-10, and $-56\%$ for GTSRB). We also implemented a timeout for each dataset and assessed the explanation size for each method under the timeout. These results are presented in Tab. 1 and demonstrate the substantial gains in explanation size achieved with our abstraction-refinement approach.

### 5.2   MINIMAL EXPLANATIONS AT DIFFERENT ABSTRACTION LEVELS

Besides improving computation time and reducing explanation size, the abstraction-refinement method allows users to observe the progressive decrease in explanation size at each abstraction level. Although networks with significant reductions initially provide larger explanations, refining these networks yields explanations of decreasing size. This transparency, from deriving abstract to refined explanations, may provide users with deeper insights into the prediction mechanism. Furthermore, it offers flexibility to halt the process when the explanation is provably sufficient, even if not provably

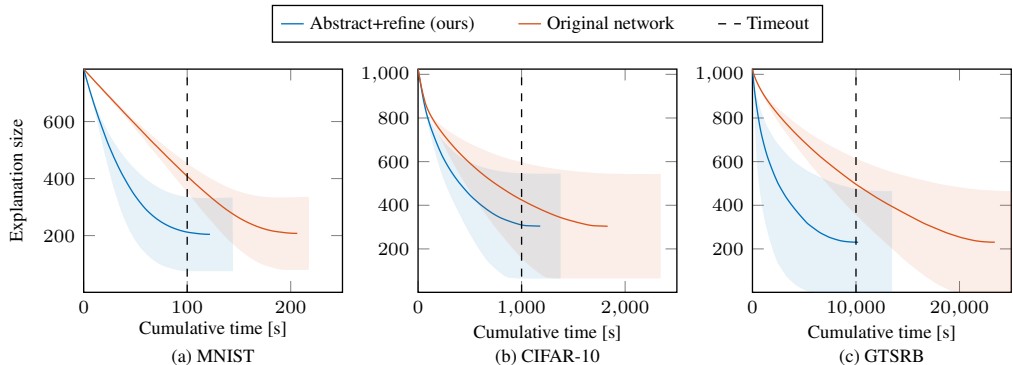

Figure 2: The explanation size over cumulative time for MNIST, CIFAR10, and GTSRB, throughout the entire abstraction-refinement algorithm, or using the standard verification algorithm on the original network. The standard deviation is shown as a shaded region.

Table 1: Mean explanation size with standard deviation using a timeout of $100s$, $1,000s$, and $10,000s$ for MNIST, CIFAR-10, and GTSRB, respectively.

| Method | Explanation size | | |
| | MNIST | CIFAR-10 | GTSRB |
| --- | --- | --- | --- |
| Abstract+refine (ours) | **204.41** $\pm$ 129.25 | **308.24** $\pm$ 236.64 | **230.60** $\pm$ 234.29 |
| Original network | 408.73 $\pm$ 36.57 | 448.68 $\pm$ 138.40 | 502.44 $\pm$ 101.69 |
| $\rho = 10\%$ | 507.33 $\pm$ 141.33 | 850.60 $\pm$ 28.86 | 502.44 $\pm$ 101.69 |
| $\rho = 20\%$ | 420.58 $\pm$ 149.36 | 806.44 $\pm$ 57.37 | 704.64 $\pm$ 168.85 |
| $\rho = 30\%$ | 340.05 $\pm$ 144.96 | 687.12 $\pm$ 130.83 | 604.96 $\pm$ 203.07 |
| $\rho = 40\%$ | 291.55 $\pm$ 126.95 | 502.52 $\pm$ 196.47 | 491.12 $\pm$ 230.24 |
| $\rho = 50\%$ | 285.47 $\pm$ 98.01 | 346.28 $\pm$ 219.37 | 392.60 $\pm$ 240.07 |
| $\rho = 60\%$ | 302.33 $\pm$ 77.50 | 314.24 $\pm$ 232.13 | 323.60 $\pm$ 237.85 |
| $\rho = 70\%$ | 325.58 $\pm$ 64.54 | 311.48 $\pm$ 233.04 | 292.64 $\pm$ 225.22 |
| $\rho = 80\%$ | 350.12 $\pm$ 53.22 | 310.24 $\pm$ 233.54 | 308.80 $\pm$ 201.25 |
| $\rho = 90\%$ | 372.69 $\pm$ 46.78 | 310.56 $\pm$ 233.08 | 333.72 $\pm$ 188.43 |

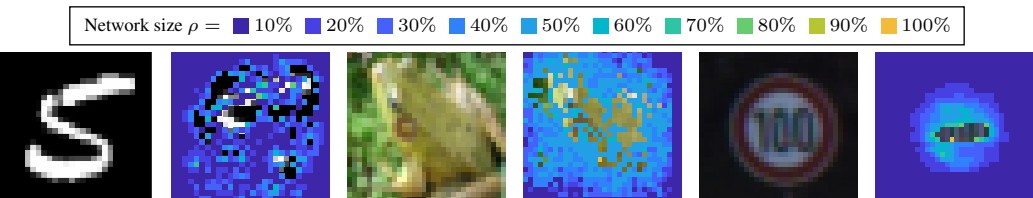

Figure 3: Examples of explanations at varying reduction rates for MNIST, CIFAR-10, and GTSRB. More examples can be found in appendix C.

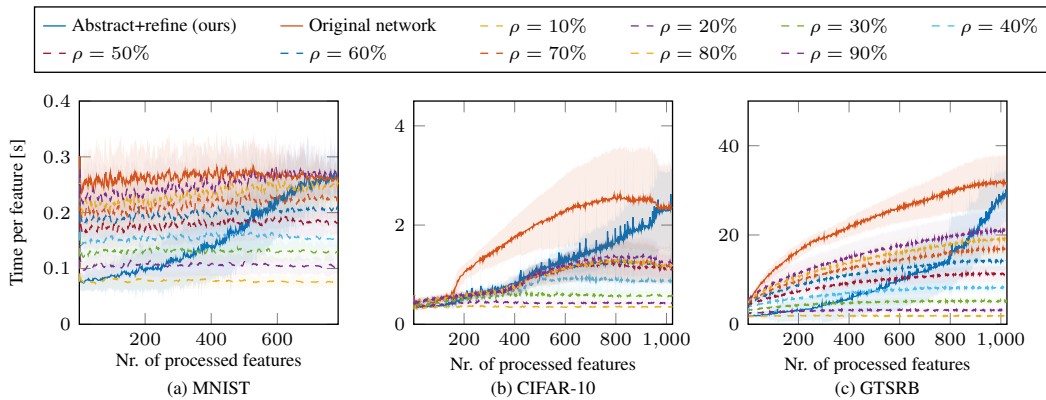

Figure 4: The percentage of processed features—identified as either included or excluded from the explanation—over cumulative time for all benchmarks, segmented by reduction rate, throughout the abstraction-refinement algorithm, or using the standard verification algorithm on the original network.

Table 2: Comparison with heuristic-based approaches, measuring sufficiency and average computation time over 100 images. Heuristic methods are faster but lack sufficiency, while our method consistently provides sufficient explanations (Prop. 2).

| Method | MNIST | | CIFAR-10 | | GTSRB | |
| | Suff. | Time | Suff. | Time | Suff. | Time |
|---|---|---|---|---|---|---|
| Anchors (Ribeiro et al., 2018) | 25% | 0.56s | 3% | 1.17s | 17% | 0.14s |
| SIS (Carter et al., 2019) | 22% | 322.72s | 0% | 553.92s | 6% | 95.13s |
| Original network | **100**% | 207.80s | **100**% | 1,838.2s | **100**% | 23,504s |
| Abstract+refine (ours) | **100**% | 121.95s | **100**% | 1,180.5s | **100**% | 10,235s |

minimal. This fine-grained process is illustrated in Fig. 3. Across all three benchmarks, small explanations can be obtained at low reduction rates (using less than 50% of the neurons).

## 5.3 EFFECT OF REDUCTION RATES

For a more detailed analysis of our findings, we present additional results on the computation of explanations at varying reduction rates within our abstraction process. In Fig. 4, we illustrate the percentage of processed features verified to be included or excluded from the explanation per reduction rate for MNIST, CIFAR-10, and GTSRB. These results highlight that the majority of the explanation processing occurs at coarser abstractions, i.e., smaller network sizes $\rho$, which accounts for the marked improvement in computation time.

## 5.4 COMPARISON TO HEURSTIC-BASED APPROACHES

In our final experiment, we compared the results of our explanations with those obtained from non-verification-based methods. Specifically, we evaluated our explanations against two widely used approaches that compute sufficient explanations: (i) Anchors (Ribeiro et al., 2018) and (ii) SIS (Carter et al., 2019). Although these methods operate relatively efficiently, they do not formally verify the sufficiency of the explanations, relying instead on a heuristic sampling across the complement. We depicted the comparisons between our verified explanations and those generated by Anchors and SIS in Tab. 2. These results highlight that while faster, these methods produce far fewer provably sufficient explanations ($\leq 25\%$).

## 6 RELATED WORK

Our work is closely related to formal explainable artificial intellicene (XAI) (Marques-Silva & Ignatiev, 2022), which aims to provide explanations with formal guarantees. Previous research has focused on deriving provable sufficient explanations for simple models like decision trees (Huang et al., 2021; Izza et al., 2022), linear models (Marques-Silva et al., 2020), monotonic classifiers (Marques-Silva et al., 2021; El Harzli et al., 2022), and tree ensembles (Izza & Marques-Silva, 2021; Ignatiev et al., 2022; Boumazouza et al., 2021). More closely related to our work are methods that derive minimal sufficient explanations for neural networks (Bassan & Katz, 2023; La Malfa et al., 2021; Wu et al., 2024). These explanations often rely on neural network verification tools, which are rapidly improving in scalability (Katz et al., 2017; Wang et al., 2021; Brix et al., 2023), though their scalability remains a key challenge, as they require executing multiple verification queries (Barceló et al., 2020; Ignatiev et al., 2019; Wu et al., 2024).

Additionally, our algorithm uses abstraction-refinement, a technique predominantly used to improve the efficiency of symbolic model checking (Clarke et al., 2000; Wang et al., 2006). This approach has also been successfully applied in software (Jhala & Majumdar, 2009; Flanagan & Qadeer, 2002), hardware verification (Andraus et al., 2007), and hybrid systems verification (Alur et al., 2000). More recently, techniques have been proposed to use abstraction-refinement by abstracting neural network sizes to improve the efficiency of certifying adversarial robustness (Elboher et al., 2020; Ladner & Althoff, 2023; Liu et al., 2024; Siddiqui et al., 2024). However, to the best of our knowledge, we are the first to adopt such an abstraction-refinement-based technique to reduce neural network sizes for providing provable explanations of neural network predictions.

## 7 LIMITATIONS AND FUTURE WORK

The primary limitation of our framework is its reliance on neural network verification queries, which currently face scalability challenges. While verification techniques are still limited in applying to state-of-the-art models, their scalability is improving rapidly (Wang et al., 2021; Brix et al., 2023). Our method adds an orthogonal step in using these tools to derive provable explanations for neural network decisions. Hence, as the scalability of verification tools improves, so will that of our approach. Additionally, our focus is on obtaining minimal sufficient explanations for predictions, but other methods exist for explaining model decisions. Future work could explore abstraction-refinement strategies for scaling various explanation types with formal guarantees by reducing the network size. If a formal explanation holds for a smaller network $f'$, it should apply to the original model $f$. For example, one could construct a smaller network where a minimal counterfactual explanation for $f'$ is also minimal for $f$ (Mothilal et al., 2020). Other approaches might study attributes of Shapley values, such as symmetry and efficiency (Sundararajan & Najmi, 2020), or properties like infidelity or consistency (Yeh et al., 2019) of feature attributions. Each property might necessitate distinct methods of abstraction between $f$ and $f'$, presenting compelling avenues for future research.

## 8 CONCLUSION

Obtaining minimal sufficient explanations for neural networks offers a promising way to provide explanations with formally verifiable guarantees. However, the scalability of generating such explanations is hindered by the need to invoke multiple neural network verification queries. Our abstraction-refinement approach addresses this by starting with a significantly smaller network and refining it as needed. This ensures that the explanations are provably sufficient for the original network and, ultimately, both provably minimal and sufficient. The method also produces intermediate explanations, allowing for an early stop when sufficient but non-minimal explanations are reached, while also offering a more fine-grained interpretation of the model prediction. Our experiments demonstrate that our approach generates minimal sufficient explanations substantially more efficiently than traditional methods, representing a significant step forward in producing explanations for neural network predictions with formal guarantees.

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

# Appendix

The appendix includes supplementary experimental results, implementation details, and proofs.

**Appendix A** contains all proofs of this paper.
**Appendix B** contains implementation details.
**Appendix C** contains supplementary results.

## A    PROOFS

In this section, we provide the missing proofs in the order they appear in the main paper.

### A.1    PROOFS OF SEC. 3: "FROM ABSTRACT NEURAL NETWORKS TO ABSTRACT EXPLANATIONS"

To prove Cor. 1, we define both a neural network and an abstract neural network layer-by-layer, as described in Sec. 3.

**Definition 6 (Neural Network)** *Let $\boldsymbol{x} \in \mathbb{R}^{n_0}$ be the input of a neural network $f$ with $\kappa$ layers, its output $\boldsymbol{y} := f(\boldsymbol{x}) \in \mathbb{R}^{n_\kappa}$ is obtained as follows:*

$$\boldsymbol{h}_0 := \boldsymbol{x}, \ \boldsymbol{h}_k := L_k\left(\boldsymbol{h}_{k-1}\right), \ \boldsymbol{y} = \boldsymbol{h}_\kappa, \quad k \in [\kappa],$$

*where $L_k \colon \mathbb{R}^{n_{k-1}} \to \mathbb{R}^{n_k}$ represents the operation of layer $k$ and is given by $L_k\left(\boldsymbol{h}_{k-1}\right) := \sigma(\boldsymbol{W}_k \boldsymbol{h}_{k-1} + \boldsymbol{b}_k)$ with weight matrix $\boldsymbol{W}_k \in \mathbb{R}^{n_k \times n_{k-1}}$, bias $\boldsymbol{b}_k \in \mathbb{R}^{n_k}$, activation function $\sigma \colon \mathbb{R}^{n_k} \to \mathbb{R}^{n_k}$, and number of neurons $n_k \in \mathbb{N}$.*

Given a set $\mathcal{S}$ and a function $f \colon \mathbb{R}^n \to \mathbb{R}^m$, we define $f(\mathcal{S}) := \{f(s) \mid s \in S\}$. An abstract network is then described by:

**Definition 7 (Abstract Neural Network)** *Let $\boldsymbol{x} \in \mathbb{R}^{n_0}$ be the input of an abstract neural network $f'$ with $\kappa$ layers, its output $\boldsymbol{y} := f'(\boldsymbol{x}) \subset \mathbb{R}^{n_\kappa}$ is obtained as follows:*

$$\mathcal{H}'_0 := \{\boldsymbol{x}\}, \ \mathcal{H}'_k := L'_k\left(\mathcal{H}'_{k-1}\right), \ \boldsymbol{y} = \mathcal{H}'_\kappa, \quad k \in [\kappa],$$

*where $L'_k \colon 2^{\left(\mathbb{R}^{n'_{k-1}}\right)} \to 2^{\left(\mathbb{R}^{n'_k}\right)}$ represents the operation of the abstract layer $k$ and is given by $L'_k\left(\mathcal{H}'_{k-1}\right) = \sigma(\boldsymbol{W}'_k \mathcal{H}'_{k-1} \oplus \boldsymbol{b}'_k)$ with weight matrix $\boldsymbol{W}'_k \in \mathbb{R}^{n'_k \times n'_{k-1}}$, bias $\boldsymbol{b}'_k \in \mathbb{R}^{n'_k}$, activation function $\sigma \colon \mathbb{R}^{n'_k} \to \mathbb{R}^{n'_k}$, number of neurons $n'_k \in \mathbb{N}$, $n'_0 := n_0, n'_\kappa := n_\kappa$, and $\oplus$ denoting the Minkowski sum.*

Let $\mathcal{X} \subset \mathbb{R}^n$ be the set of points satisfying the input specification $\psi_{\text{in}}(\boldsymbol{x})$ for a point $\boldsymbol{x} \in \mathbb{R}^n$ (Sec. 2.1). As mentioned in Sec. 2.1, the exact output $\mathcal{Y}^* := f(\mathcal{X})$ is often infeasible to compute, and an enclosure $\mathcal{Y} \supset \mathcal{Y}^*$ is computed instead by bounding all approximation errors. This is realized by iteratively propagating $\mathcal{X}$ through all layers and enclosing the output of each layer. For example, given an input set $\mathcal{H}_{k-1} \subset \mathcal{H}^*_{k-1}$ to layer $k$, we obtain the output $\mathcal{H}_k \supset L_k\left(\mathcal{H}_{k-1}\right) = \mathcal{H}^*_k$, with $\mathcal{H}^*_0 = \mathcal{H}_0 = \mathcal{X}$ and $\mathcal{Y} = \mathcal{H}_\kappa$. Let $\mathcal{H}_k, \mathcal{Y}$ and $\mathcal{H}'_k, \mathcal{Y}'$ denote the enclosures of the sets $\mathcal{H}^*_k, \mathcal{Y}_k$ using the original network $f$ and the abstract network $f'$, respectively. In this work, we use a neuron-merging construction defined by Ladner & Althoff (2023):

**Proposition 3 (Neuron-Merging Construction (Ladner & Althoff, 2023, Prop. 4))** *Given a layer $k \in [\kappa - 1]$ of a network $f$, output bounds $\mathcal{I}_k \subset \mathbb{R}^{n_k}$, a set of neurons to merge $\mathcal{B}_k \subseteq [n_k]$, and the indices of the remaining neurons $\overline{\mathcal{B}}_k := [n_k] \backslash \mathcal{B}_k$, the layer $k$ and $k+1$ are constructed as follows:*

$$\boldsymbol{W}'_k := \boldsymbol{W}_{k(\overline{\mathcal{B}}_k, \cdot)}, \ \boldsymbol{b}'_k := \boldsymbol{b}_{k(\overline{\mathcal{B}}_k)}, \quad \boldsymbol{W}'_{k+1} = \boldsymbol{W}_{k+1(\cdot, \overline{\mathcal{B}}_k)}, \ \boldsymbol{b}'_{k+1} = \boldsymbol{b}_{k+1} \oplus \boldsymbol{W}_{k+1(\cdot, \mathcal{B}_k)} \mathcal{I}_{k(\mathcal{B}_k)},$$

*where $\boldsymbol{W}_{k+1(\cdot, \mathcal{B}_k)} \mathcal{I}_{k(\mathcal{B}_k)}$ is the approximation error. The construction ensures that $\mathcal{H}^*_{k+1} \subseteq \mathcal{H}'_{k+1}$.*

Given a neural network $f$, an input $\boldsymbol{x}$, a perturbation radius $\epsilon_p$, and a subset $\mathcal{S} \subseteq [n]$, we say that $f'$ is an abstract network constructed by neuron merging with respect to the query $\langle f, \boldsymbol{x}, \mathcal{S}, \epsilon_p \rangle$ if we define the input set $\mathcal{X} := \mathcal{B}_p^{\epsilon_p}(\boldsymbol{x}_\mathcal{S}; \tilde{\boldsymbol{x}}_{\bar{\mathcal{S}}})$ and recursively apply the neuron-merging construction as described in 3 for any two layers $L_{k-1}$ and $L_k$. We can now provide an explicit proof of our corollary:

**Corollary 1 (Explanation Under Abstraction)** *Given a neural network $f$, an input $\boldsymbol{x}$, a perturbation radius $\epsilon_p$, a subset $\mathcal{S} \subseteq [n]$, let $f'$ be an abstract network constructed by neuron merging concerning the query $\langle f, \boldsymbol{x}, \mathcal{S}, \epsilon_p \rangle$. Then, it holds that:*

$$\text{suff}(f', \boldsymbol{x}, \mathcal{S}, \epsilon_p) \implies \text{suff}(f, \boldsymbol{x}, \mathcal{S}, \epsilon_p).$$

*Proof.* We prove this statement by contradiction: Assume that $\mathcal{S}$ is a sufficient explanation for the abstract network, i.e., for $\langle f', \boldsymbol{x}, \epsilon_p \rangle$, but not for the original network, i.e., for $\langle f, \boldsymbol{x}, \epsilon_p \rangle$. Given that $\mathcal{S}$ is a sufficient explanation for $\langle f', \boldsymbol{x}, \epsilon_p \rangle$, the following holds (Def. 3):

$$\forall j \neq t \in [c], \ \forall \tilde{\boldsymbol{x}} \in B_p^{\epsilon_p}(\boldsymbol{x}): \quad [\min(f'(\boldsymbol{x}_\mathcal{S}; \tilde{\boldsymbol{x}}_{\bar{\mathcal{S}}})_{(t)}) \geq \max(f'(\boldsymbol{x}_\mathcal{S}; \tilde{\boldsymbol{x}}_{\bar{\mathcal{S}}})_{(j)})], \tag{1}$$

where $t := \arg\max_j \ f(\boldsymbol{x})_{(j)}$ is the target class. Moreover, since $\mathcal{S}$ is *not* a suffcient explanation concerning $\langle f, \boldsymbol{x}, \epsilon_p \rangle$ it follows that (Def. 1):

$$\exists \tilde{\boldsymbol{x}}' \in B_p^{\epsilon_p}(\boldsymbol{x}): \quad [\arg\max_j \ f(\boldsymbol{x}_\mathcal{S}; \tilde{\boldsymbol{x}}'_{\bar{\mathcal{S}}})_{(j)} \neq \arg\max_j \ f(\boldsymbol{x})_{(j)} = t]. \tag{2}$$

Since Eq. (1) is valid for *any* $\tilde{\boldsymbol{x}} \in B_p^{\epsilon_p}(\boldsymbol{x})$, it explicitly applies to $\tilde{\boldsymbol{x}}' \in B_p^{\epsilon_p}(\boldsymbol{x})$ as well. Specifically, we have:

$$\forall j \in [c]\backslash\{t\}, \quad [\min(f'(\boldsymbol{x}_\mathcal{S}; \tilde{\boldsymbol{x}}'_{\bar{\mathcal{S}}})_{(t)}) \geq \max(f'(\boldsymbol{x}_\mathcal{S}; \tilde{\boldsymbol{x}}'_{\bar{\mathcal{S}}})_{(j)})]. \tag{3}$$

We now assert that to establish the correctness of the corollary, it suffices to demonstrate that:

$$f(\boldsymbol{x}_\mathcal{S}; \tilde{\boldsymbol{x}}'_{\bar{\mathcal{S}}}) \in f'(\boldsymbol{x}_\mathcal{S}; \tilde{\boldsymbol{x}}'_{\bar{\mathcal{S}}}) \tag{4}$$

The rationale is as follows: if Eq. (4) holds, it would directly contradict our initial assumption. To begin, observe that Eq. (4) directly leads to:

$$\forall j \in [c], \quad [\min(f'(\boldsymbol{x}_\mathcal{S}; \tilde{\boldsymbol{x}}'_{\bar{\mathcal{S}}})_{(j)}) \leq f(\boldsymbol{x}_\mathcal{S}; \tilde{\boldsymbol{x}}'_{\bar{\mathcal{S}}})_{(j)} \leq \max(f'(\boldsymbol{x}_\mathcal{S}; \tilde{\boldsymbol{x}}'_{\bar{\mathcal{S}}})_{(j)})], \tag{5}$$

and therefore, based on Eq. (3), it will directly follow that:

$$\forall j \in [c]\backslash\{t\}, \quad [f(\boldsymbol{x}_\mathcal{S}; \tilde{\boldsymbol{x}}'_{\bar{\mathcal{S}}})_{(t)} \geq \min(f'(\boldsymbol{x}_\mathcal{S}; \tilde{\boldsymbol{x}}'_{\bar{\mathcal{S}}})_{(t)}) \geq \max(f'(\boldsymbol{x}_\mathcal{S}; \tilde{\boldsymbol{x}}'_{\bar{\mathcal{S}}})_{(j)}) \geq f(\boldsymbol{x}_\mathcal{S}; \tilde{\boldsymbol{x}}'_{\bar{\mathcal{S}}})_{(j)}]. \tag{6}$$

Now, specifically, given that the following condition is satisfied:

$$\forall j \neq t \in [c], \quad [f(\boldsymbol{x}_\mathcal{S}; \tilde{\boldsymbol{x}}'_{\bar{\mathcal{S}}})_{(t)} \geq f(\boldsymbol{x}_\mathcal{S}; \tilde{\boldsymbol{x}}'_{\bar{\mathcal{S}}})_{(j)}]. \tag{7}$$

This indicates that $\arg\max_j \ f(\boldsymbol{x}_\mathcal{S}; \tilde{\boldsymbol{x}}'_{\bar{\mathcal{S}}})(j) = t$, which contradicts the assumption stated in Eq. (2).

We are now left to prove that $f(\boldsymbol{x}_\mathcal{S}; \tilde{\boldsymbol{x}}'_{\bar{\mathcal{S}}}) \in f'(\boldsymbol{x}_\mathcal{S}; \tilde{\boldsymbol{x}}'_{\bar{\mathcal{S}}})$. Let $\boldsymbol{h}_k$ and $\mathcal{H}'_k$ be as defined in Def. 6 and Def. 7, respectively, for the input $(\boldsymbol{x}_\mathcal{S}; \tilde{\boldsymbol{x}}'_{\bar{\mathcal{S}}}) \in \mathcal{X}$, where $\mathcal{X} := B_p^{\epsilon_p}(\boldsymbol{x}_\mathcal{S}; \tilde{\boldsymbol{x}}'_{\bar{\mathcal{S}}})$. Recall that this is defined since the merging is performed with respect to the query $\langle f, \boldsymbol{x}, \mathcal{S}, \epsilon_p \rangle$. We show by induction that the statement $f(\boldsymbol{x}_\mathcal{S}; \tilde{\boldsymbol{x}}'_{\bar{\mathcal{S}}}) \in f'(\boldsymbol{x}_\mathcal{S}; \tilde{\boldsymbol{x}}'_{\bar{\mathcal{S}}})$ holds:

*Induction hypothesis.* $k \in [\kappa]$: The condition $\boldsymbol{h}_k \in \mathcal{H}'_k$ is satisfied if a neuron merging operation was performed between any two layers up to and including layer $k - 1$.

*Induction base.* $k = 0$: Trivially holds (Def. 7).

*Induction step.* $k \to k+1$: We need to show that $\boldsymbol{h}_{k+1} \in \mathcal{H}'_{k+1}$. Let $\mathcal{B}_k, \mathcal{I}_k$ be as in Prop. 3 and $\mathcal{H}'_k$ the output set of layer $k$ before merging. Thus, $\mathcal{I}_k \supset \mathcal{H}'_k$ holds. From the induction hypothesis, we know that $\boldsymbol{h}_k \in \mathcal{H}'_k$ holds. Recall from Def. 6 that $\boldsymbol{h}_{k+1} = \sigma(\boldsymbol{W}_{k+1}\boldsymbol{h}_k + \boldsymbol{b}_{k+1})$. Doing the same for $\mathcal{H}'_k$ (Def. 7) and applying the neuron merging construction (Prop. 3) gives us:

$$
\begin{aligned}
\boldsymbol{h}_{k+1} \in \sigma(\boldsymbol{W}_{k+1}\mathcal{H}'_k \oplus \boldsymbol{b}_{k+1}) &= \sigma(\boldsymbol{W}_{k+1(\cdot,\mathcal{B}_k)}\mathcal{H}'_{k(\mathcal{B}_k)} \oplus \boldsymbol{W}_{k+1(\cdot,\overline{\mathcal{B}}_k)}\mathcal{H}'_{k(\overline{\mathcal{B}}_k)} \oplus \boldsymbol{b}_{k+1}) \\
&\subseteq \sigma(\boldsymbol{W}_{k+1(\cdot,\mathcal{B}_k)}\mathcal{I}_{k(\mathcal{B}_k)} \oplus \boldsymbol{W}_{k+1(\cdot,\overline{\mathcal{B}}_k)}\mathcal{H}'_{k(\overline{\mathcal{B}}_k)} \oplus \boldsymbol{b}_{k+1}) \\
&= \mathcal{H}'_{k+1},
\end{aligned}
$$

which proves the induction step. As the induction hypothesis holds for $k = \kappa$, we conclude that $f'(\boldsymbol{x}_\mathcal{S}; \tilde{\boldsymbol{x}}'_{\overline{\mathcal{S}}}) \in f'(\boldsymbol{x}_\mathcal{S}; \tilde{\boldsymbol{x}}'_{\overline{\mathcal{S}}})$ must be true. This, as previously explained, implies that $\arg\max_j f(\boldsymbol{x}_\mathcal{S}; \tilde{\boldsymbol{x}}'_{\overline{\mathcal{S}}})_{(j)} = t$, which contradicts our assumption in Eq. (2).

$\square$

### A.2 PROOFS OF SEC. 4: "FROM REFINING NEURAL NETWORKS TO REFINING EXPLANATIONS"

**Proposition 1 (Refined Abstract Network)** *Given a neural network $f$, an input $\boldsymbol{x}$, a perturbation radius $\epsilon_p \in \mathbb{R}$, a subset $\mathcal{S} \subseteq [n]$, and an abstract network $f'$ with reduction rate $\rho' \in [0,1]$, we can construct a refined abstract network $f''$ from $f, f'$ with reduction rate $\rho'' > \rho'$, for which holds that:*

$$
\forall \tilde{\boldsymbol{x}} \in B_p^{\epsilon_p}(\boldsymbol{x}): f(\boldsymbol{x}_\mathcal{S}; \tilde{\boldsymbol{x}}_{\overline{\mathcal{S}}}) \in f''(\boldsymbol{x}_\mathcal{S}; \tilde{\boldsymbol{x}}_{\overline{\mathcal{S}}}) \subset f'(\boldsymbol{x}_\mathcal{S}; \tilde{\boldsymbol{x}}_{\overline{\mathcal{S}}}).
$$

*Proof.* The containment of $f(\boldsymbol{x}_\mathcal{S}; \tilde{\boldsymbol{x}}_{\overline{\mathcal{S}}})$ follows using an analogous proof as for Cor. 1. While the subset relation does not hold in general when applying the abstraction (Prop. 3) using $\rho''$ instead of $\rho'$ as different neurons might be merged, one can restrict the neurons that are allowed to be merged to the subset of neurons $\mathcal{N}' = \cup_{k \in [\kappa-1]}\mathcal{B}_k$ that were merged to obtain $f'$. Using this restriction and as $\rho'' > \rho'$ holds, $\mathcal{N}'' \subset \mathcal{N}'$ holds. Then, as all additionally merged neurons $\mathcal{N}' \setminus \mathcal{N}''$ in $f'$ induce outer approximations and everything else is equal, the subset relation holds. $\square$

**Corollary 2 (Intermediate Sufficient Explanation)** *Let there be a neural network $f$, an abstract network $f'$, and a refined neural network $f''$. Then, it holds that:*

$$
\begin{aligned}
\text{suff}(f', \boldsymbol{x}, \mathcal{S}, \epsilon_p) &\implies \text{suff}(f'', \boldsymbol{x}, \mathcal{S}, \epsilon_p) \text{ and} \\
\text{suff}(f'', \boldsymbol{x}, \mathcal{S}, \epsilon_p) &\implies \text{suff}(f, \boldsymbol{x}, \mathcal{S}, \epsilon_p).
\end{aligned}
$$

*Proof.* We show the first implication by contradiction: Let us assume that $\mathcal{S}$ is a sufficient explanation for $f'$ but not for $f''$. Thus, the query $\langle f'', \boldsymbol{x}, \mathcal{S}, \epsilon_p \rangle$ does not fulfill Def. 3. However, as $f''(\boldsymbol{x}_\mathcal{S}; \tilde{\boldsymbol{x}}_{\overline{\mathcal{S}}}) \subset f'(\boldsymbol{x}_\mathcal{S}; \tilde{\boldsymbol{x}}_{\overline{\mathcal{S}}})$ holds due to Prop. 1, the query $\langle f', \boldsymbol{x}, \mathcal{S}, \epsilon_p \rangle$ can also not fulfill Def. 3, which contradicts our assumption. The proof for the second implication is analogous. $\square$

**Corollary 3 (Intermediate Minimal Sufficient Explanation)** *Let there be a neural network $f$, an abstract network $f'$, and a refined neural network $f''$. Then, if $\mathcal{S}$ is a sufficient explanation concerning $f$, $f'$, and $f''$, it holds that:*

$$
\begin{aligned}
\text{min-suff}(f, \boldsymbol{x}, \mathcal{S}, \epsilon_p) &\implies \text{min-suff}(f'', \boldsymbol{x}, \mathcal{S}, \epsilon_p) \text{ and} \\
\text{min-suff}(f'', \boldsymbol{x}, \mathcal{S}, \epsilon_p) &\implies \text{min-suff}(f', \boldsymbol{x}, \mathcal{S}, \epsilon_p).
\end{aligned}
$$

*Proof.* The statement is shown by contradiction for the relation of $f$ and $f''$: Let us assume that the explanation $\mathcal{S}$ is minimal for $f$ but not for $f''$. Thus, there must be a $\mathcal{S}' \subset \mathcal{S}$ which is minimal for $f''$. However, this cannot be an explanation for $f$ as $\mathcal{S}$ is already minimal for $f$, From Cor. 1 follows that $\mathcal{S}'$ is also a minimal explanation for $f$, which contradicts our assumption that $\mathcal{S}$ is already minimal. Analogous holds for $f''$ and $f'$. $\square$

**Proposition 2 (Greedy Minimal Sufficient Explanation Search)** *Alg. 2 produces a provably sufficient and minimal explanation $\mathcal{S}$ concerning the query $\langle f, \boldsymbol{x}, \mathcal{S}, \epsilon_p \rangle$, which converges to the same explanation as obtained by Alg. 1.*

*Proof.* All line numbers are with respect to Alg. 2: The invariant described in line 3 holds due to Cor. 2. Thus, the final explanation $\mathcal{S}$ is sufficient concerning the original network $f$.

We show the minimality by contradiction: Let us assume the final explanation $\mathcal{S}$ is not minimal: There exists a feature $i \in [n]$ such that $\mathcal{S} \setminus \{i\}$ is still sufficient (Def. 2). It follows that we cannot have found a counterexample in lines 10 to 10, Thus, to break the do-while loop, the sufficiency check in line 6 must have passed either on an abstract network or eventually after all refinement steps on the original network. However, this would remove feature $i$ from the explanation, which violates our assumption.

We converge to the same explanation as Alg. 1, as we process the features in the same order, Cor. 2 holds, and if removing a feature results in a non-sufficient explanation and no counterexample on the original network can be found, we refine the abstract networks until we converge to the original network. □

# B  IMPLEMENTATION DETAILS

In this section, we offer further technical details about the implementation of our algorithms and provide specifics on the model architectures and training methods used in this study.

## B.1  IMPLEMENTATION DETAILS OF ALGORITHM 1 AND ALGORITHM 2

Both Alg. 1 and Alg. 2 iterate over the features in a specified order. This approach aligns with the methodologies used in (Wu et al. (2024); Izza et al. (2024); Bassan & Katz (2023)), where a sensitivity traversal over the features is employed. We prioritize iterating over features with the lowest sensitivity first, as they are most likely to be successfully freed, thus leading to a smaller explanation. As we refine the abstract network following Prop. 1, we define a series of reduction rates used during each refinement step. For simplicity, we start with a coarsest abstraction at a reduction rate $\rho = 10\%$ of the original network's neurons and increase $\rho$ by $10\%$ at each subsequent refinement, until $\rho = 100\%$ is reached, which restores the original network.

We also note that while MNIST utilizes grayscale images, both CIFAR-10 and GTSRB use RGB images. Following standard practices (Wu et al. (2024); Ribeiro et al. (2018); Carter et al. (2019; 2021)) for colored images, we provide explanations for CIFAR-10 and GTSRB on a per-pixel basis, rather than at the neuron level; this means we either include/exclude all color channels of a pixel within the explanation or none. Consequently, the maximum size of an explanation, $|\mathcal{S}|$, is $32 \cdot 32$ instead of $32 \cdot 32 \cdot 3$. For MNIST, which has only one color channel, the maximum size is always $|\mathcal{S}| = 28 \cdot 28$.

## B.2  TRAINING AND MODEL IMPLEMENTATION

For MNIST and CIFAR-10, we utilized common models from the neural network verification competition (VNN-COMP) (Brix et al., 2023), which are frequently used in experiments related to neural network verification tasks. Specifically, the MNIST model architecture is sourced from the ERAN benchmark within VNN-COMP, and the CIFAR-10 model is derived from the "marabou" benchmark. Since GTSRB is not directly utilized in VNN-COMP, we trained this model using a batch size of 32 for 10 epochs with the ADAM optimizer, achieving an accuracy of $84.8\%$. The precise dimensions and configurations of the models used for both VNN-COMP (MNIST and CIFAR-10) and GTSRB are provided: Table 3 for MNIST, Table B.2, and Table B.2 for GTSRB. For MNIST and GTSRB, we use a perturbation radius $\epsilon_{\infty} = 0.01$ as commonly used in VNN-COMP benchmarks, and for CIFAR-10, we use a smaller perturbation radius $\epsilon_{\infty} = 0.001$ as we have found this network to be not very robust.

Table 3: Dimensions for the MNIST classifier.

| Layer Type | Paramater | Activation |
|---|---|---|
| Input | $784 \times 200$ | Sigmoid |
| Fully-Connected (6 layers) | $200 \times 200$ | Sigmoid |
| Fully-Connected | $200 \times 10$ | Softmax |

Table 4: Dimensions for the CIFAR-10 classifier.

| Layer Type | Paramater | Activation |
|---|---|---|
| Input | $32 \times 32 \times 3$ | ReLU |
| Convolution | $32 \times 3 \times 4 \times 4$ | ReLU |
| Convolution | $64 \times 32 \times 4 \times 4$ | ReLU |
| Fully-Connected | $32768 \times 128$ | ReLU |
| Fully-Connected | $128 \times 64$ | ReLU |
| Fully-Connected | $64 \times 10$ | Softmax |

Table 5: Dimensions for the GTSRB classifier.

| Layer Type | Paramater | Activation |
|---|---|---|
| Input | $32 \times 32 \times 3$ | Sigmoid |
| Convolution | $16 \times 3 \times 3 \times 3$ | Sigmoid |
| AveragePool | $2 \times 2$ | — |
| Convolution | $32 \times 16 \times 3 \times 3$ | Sigmoid |
| AveragePool | $2 \times 2$ | — |
| Fully-Connected | $4608 \times 128$ | Sigmoid |
| Fully-Connected | $128 \times 43$ | softmax |

## C  SUPPLEMENTRY RESULTS

In this section, we present further experimental results to complement those discussed earlier. We begin by expanding on Fig. 2, which illustrates the change in explanation size over time for the standard verification method versus the abstraction-refinement approach. In Fig. 5, we offer a similar comparison, this time focusing on the number of processed features, i.e., features that have been selected to be included or excluded from the explanation. It is evident that the abstraction-refinement method processes features more efficiently than the standard approach, leading to enhanced scalability.

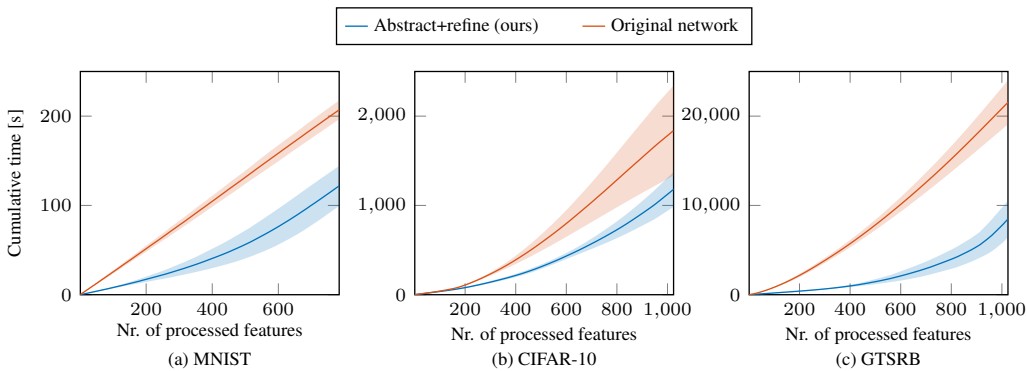

(a) MNIST  (b) CIFAR-10  (c) GTSRB

Figure 5: The percentage of features successfully processed—identified as either included or excluded from the explanation—over cumulative time for MNIST, CIFAR10, and GTSRB, throughout the entire abstraction-refinement algorithm, or using the standard verification algorithm on the original network. The standard deviation is shown as shaded region.

We continue to build on the findings presented in Fig. 4, which illustrates the number of features processed at various reduction rates. In Fig. 6, we similarly demonstrate the change in explanation size over time across different reduction rates. As lower reduction rates $\rho$ initially have a much steeper curve than larger ones; thus, the explanation size is reduced faster. However, these lower reduction rates converge to higher explanation sizes than larger reduction rates. Our approach benefits from both worlds by initially using the steepest curve to reduce the explanation size, and automatically switching to the next steeper curve if no features can be freed anymore using the current rate.

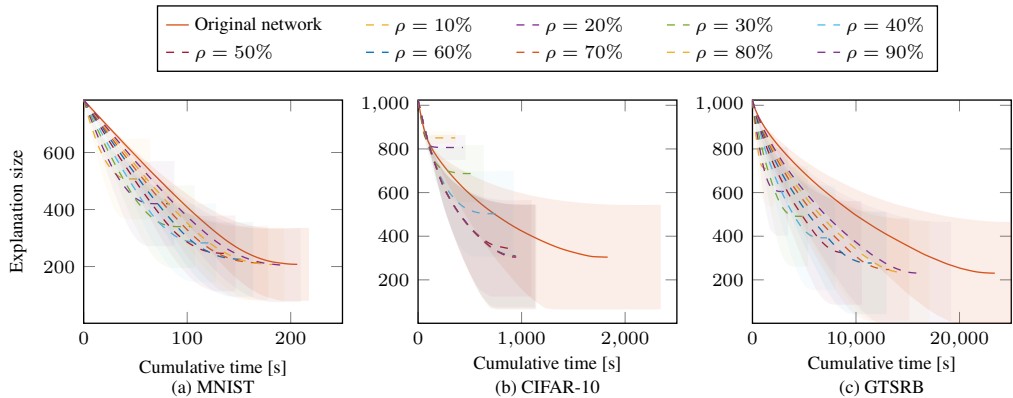

Figure 6: The explanation size over cumulative time for MNIST, CIFAR10, and GTSRB, segmented by reduction rate, throughout the entire abstraction-refinement algorithm, or using the standard verification algorithm on the original network. The standard deviation is shown as shaded region.

Lastly, we provide extra figures that depict the iterative abstraction-refinement process in the explanations across different reduction rates. Fig. 7 displays the initial images paired with a colored grid, where each color represents a specific reduction rate. These images are selected from the MNIST, CIFAR10, and GTSRB datasets. In the last row, we show some explanations on GTSRB images with unexpected explanations. For example, for the first image in the last row, the red circle surrounding the sign does not seem to be very important, as these pixels could be removed from the explanations using the coarsest abstraction ($\rho = 10\%$).

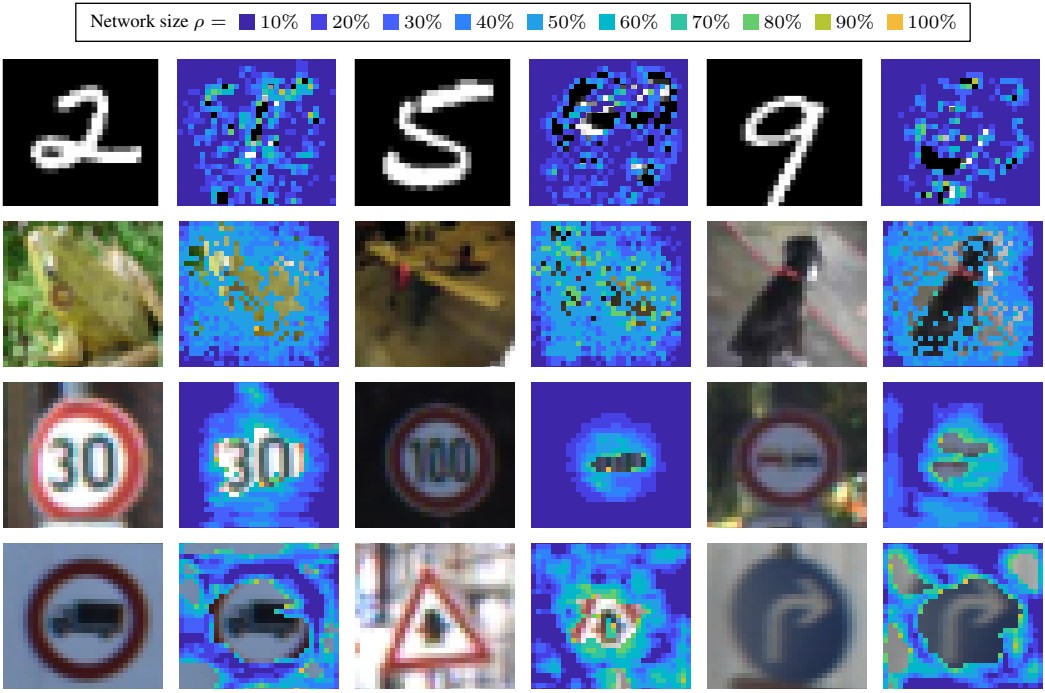

Figure 7: Original images compared to images featuring the complete abstraction-refinement grid at various abstraction rates for MNIST, CIFAR10, and GTSRB.

Additionally, to provide a more detailed visualization of the entire abstraction-refinement explanation process, which allows users to halt the verification at any stage, we include visualizations of both

abstract and refined explanations at various steps and reduction rates. These visualizations are shown for all three benchmarks — MNIST, CIFAR-10, and GTSRB — in Fig. 8.

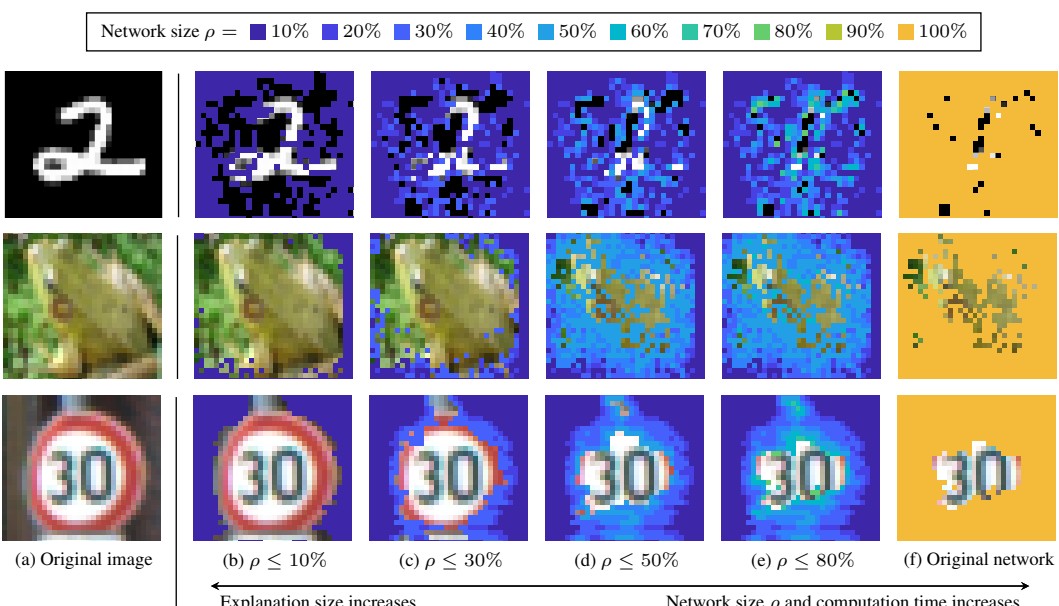

Network size $\rho =$ ■10% ■20% ■30% ■40% ■50% ■60% ■70% ■80% ■90% ■100%

| (a) Original image | (b) $\rho \leq 10\%$ | (c) $\rho \leq 30\%$ | (d) $\rho \leq 50\%$ | (e) $\rho \leq 80\%$ | (f) Original network |

Explanation size increases      Network size $\rho$ and computation time increases

Figure 8: A step-by-step visualization of the different abstraction levels for both the network and explanation across MNIST, CIFAR-10, and GTSRB.

# D  ADDITIONAL EXPERIMENTS AND ABLATIONS

## D.1  SUFFICIENCY-COMPUTATION TIME TRADE-OFF

In this subsection, we will examine the impact of varying perturbation radii $\epsilon_p$ on our experimental results. Larger $\epsilon_p$ perturbations make each query more challenging but provide stronger sufficiency guarantees. However, as the sufficiency conditions become more stringent, the total number of queries decreases, leading to larger explanation sizes. We conducted an experiment on MNIST using different perturbation radiuses, with the results presented in Tab. 6.

Table 6: Impact of perturbation radius $\epsilon_p$ on explanation size and computation size.

| Perturbation Radius | Explanation Size | Computation Time |
|---|---|---|
| 0.012 | 219.450±142.228 | 110.111±33.712 |
| 0.011 | 186.970±140.435 | 101.881±41.625 |
| 0.010 | 153.240±135.733 | 94.897±46.244 |
| 0.009 | 119.040±127.271 | 81.889±52.578 |
| 0.008 | 87.530±113.824 | 62.607±58.084 |
| 0.007 | 59.420±95.607 | 53.072±56.709 |

## D.2  FEATURE ORDERING

We illustrate the impact of different feature orderings on the explanations generated by our method. While we adopt the approach proposed by Wu et al. (2024), which orders features by descending sensitivity values, we also present results for explanation sizes and computation times using alternative feature orderings in our MNIST configuration. These alternatives include ordering by descending Shapley value attributions (Lundberg & Lee, 2017) and, for comparison, a straightforward in-order traversal that results in larger subsets. The results are summarized in Tab. 7.

Table 7: Impact of feature order on computation time and explanation size.

| Method | Feature Order | Computation Time | Explanation Size |
|---|---|---|---|
| Ours | Sensitivity | 90.26±44.54 | 153.24±135.73 |
| Ours | Shapley | 93.10±45.39 | 175.70±150.09 |
| Ours | In-order | 98.10±46.42 | 231.46±160.73 |

## D.3  EXTENSION TO ADDITIONAL DOMAINS

Although our method primarily targets classification tasks in image domains, it is model-type agnostic. Furthermore, it can be easily adapted to regression tasks by defining the sufficiency conditions for a subset $\mathcal{S}$ for a model $f : \mathbb{R}^n \to \mathbb{R}$ and some given input $\boldsymbol{x} \in \mathbb{R}^n$ as:

$$\forall \tilde{\boldsymbol{x}} \in B_p^{\epsilon_p}(\boldsymbol{x})\colon \quad \| f(\boldsymbol{x}_{\mathcal{S}}; \tilde{\boldsymbol{x}}_{\bar{\mathcal{S}}}) - f(\boldsymbol{x})\|_p \leq \delta, \quad \delta \in \mathbb{R}_+.$$

**Comparison to results over Taxinet ((Wu et al., 2024))**: We aimed to compare our results over regression tasks to those conducted by Wu et al. (2024) which ran a "traditional" computation of a provably sufficient explanation for neural networks over the Taxinet benchmark, which is a real-world safety-critical airborne navigation system (Julian et al., 2020). The authors of Wu et al. (2024) obtain minimal sufficient explanations over three different benchmarks of varying sizes, two of which are relatively small, and one which is larger (the CNN architecture). We performed experiments using this architecture. Our abstraction refinement approach obtained explanations within $35.71 \pm 3.71$ seconds and obtained explanations of size $699.30 \pm 169.34$, which provides a substantial improvement over the results reported by Wu et al. (2024) (8814.85 seconds, and explanation size was not reported). We additionally provide visualizations for some of our obtained explanations (Fig. 9 and 10).

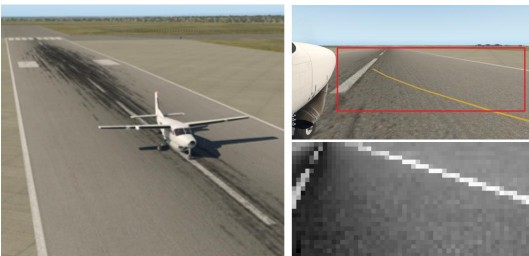

Figure 9: An autonomous aircraft taxiing scenario (Julian et al., 2020), where images captured by a camera mounted on the right wing are cropped (red box) and downsampled

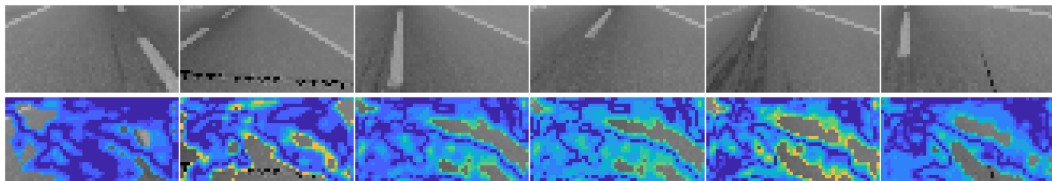

Figure 10: Varying results of explanations across different abstraction levels for the Taxinet benchmark.

**Extension to language tasks.** We present results from experiments conducted on the safeNLP benchmark Casadio et al. (2024), trained on the medical safety NLP dataset sourced from the annual neural network verification competition (VNN-COMP) (Brix et al., 2023). Notably, this benchmark is the only language-domain dataset included in the competition. The $\epsilon$ perturbations are applied within a latent space that represents an embedding of the input, thereby ensuring that the perturbations preserve the meaning of the sentence. Our findings are as follows: the traditional (non-abstraction-refinement) approach executed in $0.71 \pm 0.24$ seconds with an explanation size of $6.67 \pm 5.06$, while the abstraction-refinement approach completed in $0.66 \pm 0.22$ seconds, achieving the same explanation size of $6.67 \pm 5.06$. The results are also visualized in Fig. 11. The performance improvement here is relatively modest, as the benchmark contains few nonlinear activations. However, as emphasized in our study and the experimental analysis, the benefits of the abstraction-refinement method become significantly more pronounced in models with a higher prevalence of nonlinear activations.

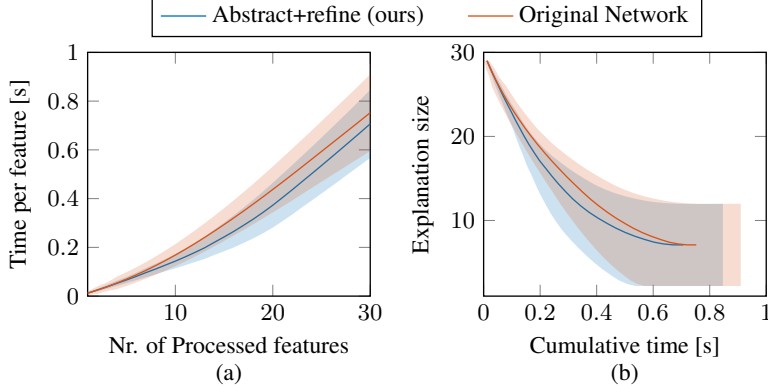

Figure 11: (a) The percentage of features successfully processed—identified as either included or excluded from the explanation—over cumulative time and (b) the explanation size over cumulative time for safeNLP, throughout the entire abstraction-refinement algorithm, or using the standard verification algorithm on the original network. The standard deviation is shown as shaded region.

