# OpenReview forum: "Abstracting and Refining Provably Sufficient Explanations of Neural Network Predictions"
_ICLR.cc/2025/Conference — Submitted to ICLR 2025_

### Official Review · Reviewer_shdo · 2024-10-18

**Soundness:** 3
**Presentation:** 4
**Contribution:** 3
**Rating:** 8
**Confidence:** 1

**Summary:**

The paper proposes to adopt an abstraction-refinement approach similar to distillation techniques for scaling up the extraction of provably sufficient explanations.
This is a relevant research direction, as traditional methods have scalability issues.
Results confirm that the extracted explanations are indeed sufficient and also minimal, and the runtime shows great improvements compared to baselines.

**Strengths:**

- Providing provably sufficient explanations is a relevant problem, and developing methods to compute them efficiently is certainly of great interest. The results confirm the effectiveness of the proposed methodology
- The overall quality of the writing and presentation is very good
- The authors provided the source code with a good description of how to use it

**Weaknesses:**

- Corollary 1 is essential for the overall message of the paper, but the proof is not self-contained and seems more like a straightforward application of the result of [1]. The authors should make the proof more self-contained, also clarifying how it builds on top of [1].

-  The title suggests that the main contribution is a method providing provably sufficient explanations for neural networks. To my understanding, however, providing a provably sufficient explanation of an abstract model as per [1] is *fairly easy*, given that a sufficient explanation for any abstract model will also be sufficient for the original one. Nonetheless, this does not guarantee the minimality of the explanation, requiring the iterative refinement method proposed in Algorithm 2. I wonder, therefore, whether the main contribution lies in providing provably sufficient explanations, or in making a provably sufficient explanation also provably minimal.






[1] Fully Automatic Neural Network Reduction for Formal Verification. Tobias Ladner and Matthias Althoff

**Questions:**

- How hard is it to extend these results to arbitrary modifications of the complement, therefore not limiting to an epsilon-ball perturbation?

---

> ### Author Response · Authors · 2024-11-18
>
> We thank the reviewer for the valuable comments. See our response below.
>
> **Corollary 1 is not entirely self-contained**
>
>
> We thank the reviewer for highlighting this point. Given the significance of Corollary 1, we agree that making the proof more self-contained, building on the work of Ladner et al., would enhance the paper's overall clarity and accessibility. We will incorporate this change in the final version.
>
>
> **Paper title implies that the contribution focuses on subset sufficiency, but proving minimality appears more challenging**
>
>
> The core idea behind proving that our abstraction-refinement strategy achieves a provably sufficient and minimal explanation is divided into two key challenges. The first challenge lies in recognizing that a neuron-merging abstraction technique, such as the one discussed by Ladner et al., can be used to abstract a neural network while ensuring that a provably sufficient explanation for the abstract model remains provably sufficient for the original model. As the reviewer rightly pointed out, the second significant challenge is applying such an abstraction method to derive explanations that are not only provably sufficient but also provably minimal with respect to the original model. While a sufficient explanation of the abstract model remains sufficient for the original model, it does not necessarily ensure provable minimality. To address this, we introduce the concept of a “provably refined network” component, enabling the gradual refinement of both the neural network and its explanation. This process is highlighted in Algorithm 2 of the paper. We agree with the reviewer that emphasizing the “minimality” aspect of our provable explanations in the paper title could enhance clarity and better reflect the contributions of the work. Thank you for bringing this to our attention.
>
>
> **Extending the results to arbitrary modifications of the complement**
>
>
> Extending the results to arbitrarily large modifications of the complement can be obtained by expanding the bounds over the input features during the certification process. Since our method is agnostic to these bounds, this extension is naturally feasible. Regarding scalability, on the one hand, each verification query becomes more computationally demanding as it involves certifying a much larger domain. On the other hand, this broader setting generates larger explanations, due to the increased likelihood of subsets failing to maintain sufficiency, and since fewer verification queries are required, the overall runtime may not increase significantly. However, it is worth mentioning that since the obtained subsets may be significantly large, the generated explanations may be less meaningful. To illustrate this, we conducted an initial experiment on MNIST, comparing perturbations within a small domain of 0.01 to those in the entire [0,1] domain. The results are as follows:
>
>
>
>
>
>
> | Perturbation Radius | Explanation Size | Generation Time |
> |---------------------|------------------|-------------------|
> |               0.01| 153.240 ± 135.733 |  94.897 ± 46.244  |
> |              [0, 1] |  768.970 ± 5.788 |  155.780 ± 2.779  |
>
>
> Building on this remark and the questions raised by reviewer t2CN, we will include an additional analysis in our paper. This analysis will evaluate our method's performance under varying $\epsilon$ perturbations across all benchmarks, offering insights into the trade-offs between sufficiency and runtime. An initial experiment on MNIST yielded the following results:
>
>
> | Perturbation Radius | Explanation Size | Computation Time |
> |---------------------|------------------|-------------------|
> |               0.012 | 219.450 ± 142.228 |  110.111 ± 33.712 |
> |               0.011 | 186.970 ± 140.435 |  101.881 ± 41.625 |
> |               0.010 | 153.240 ± 135.733 |  94.897 ± 46.244  |
> |               0.009 | 119.040 ± 127.271 |  81.889 ± 52.578  |
> |               0.008 | 87.530 ± 113.824 |  62.607 ± 58.084  |
> |               0.007 |  59.420 ± 95.607 |  53.072 ± 56.709  |
>
>
> In the final version, we will include the complete experiment across all benchmarks. Thank you for highlighting this point.

---

> ### Comment · Reviewer_shdo · 2024-11-19
> **Feedback on Author Response**
>
> Thank you for your answer. I appreciated your new experiments on the increasing radius. I have one final doubt regarding your second answer. I would appreciate it if you could clarify the following point.
>
> > [...] The first challenge lies in recognizing that a neuron-merging abstraction technique, such as the one discussed by Ladner et al., can be used to abstract a neural network while ensuring that a provably sufficient explanation for the abstract model remains provably sufficient for the original model [...]
>
> What do you mean, practically, by "recognizing"? It seems more of a motivation behind the paper's idea rather than a key contribution to the paper. My concern is further supported by the fact that Corollary 1 appeared more like a straightforward application of the result of [1], without highlighting the contribution provided by the authors. Maybe, uploading your revised version of Corollary 1, where you mentioned that you clarified this, would help.
>
> Thank you again for your answer.

---

> > ### Author Response · Authors · 2024-11-21
> >
> > Thank you for your response. We have included a complete proof for Cor. 1 in the appendix as requested. We also appreciate your suggestion to rigorously prove this in greater detail, as it enhances the clarity and presentation of this aspect of the paper.
> >
> >
> > By "recognizing," we refer to the following: First, recall that an abstract neural network, as defined in the paper, differs from a neural network $f := \mathbb{R}^n \to \mathbb{R}^c$ and is defined as $f' := \mathbb{R}^n \to 2^{(\mathbb{R}^c)}$, where $c$ represents the number of classes. Definition 3 in our paper formalizes the concept of a sufficient explanation for an abstract neural network $f'$ (an “abstract explanation”). We define this formally as a subset $\mathcal{S}\subseteq \\{1,\ldots,n\\}$ for which it holds that:
> >
> > $$
> > \begin{aligned}
> > \forall j\neq t \in [c] , \  \forall \mathbf{\tilde{x}}\in\mathcal{B}_p^{\epsilon_p}(\mathbf{x}): \quad [\text{min}(f’(\mathbf{x}\_{\mathcal{S}};\mathbf{\tilde{x}}\_{\bar{\mathcal{S}}})\_{(t)})\geq \text{max}(f’(\mathbf{x}\_{\mathcal{S}};\mathbf{\tilde{x}}\_{\bar{\mathcal{S}}})\_{(j)})], \newline
> > \text{such} \ \ \text{that} \ \ \mathbf{B}\_p^{\epsilon\_p}(\mathbf{x}):=\\{\mathbf{\tilde{x}}\in\mathbb{R}^n \ | \ ||\mathbf{x}-\mathbf{\tilde{x}}||\_p\leq \epsilon\_p\\}.
> > \end{aligned}
> > $$
> >
> > The idea of this construction is that since the image of $f’$ is defined over continuous sets (in contrast to the image of $f$ which is in $\mathbb{R}^n$), we can impose a stronger constraint by ensuring that the minimum achievable value for the target class exceeds the maximum value of all other classes.
> >
> > By introducing this new concept, we can then show that under certain conditions, any subset $\mathcal{S}$ meeting the criteria for a provably sufficient explanation for the abstracted model $f’$ under this definition also qualifies as a sufficient explanation for the original model $f$. This conclusion holds, for example, when the output sets of $f’$ are constructed by merging neurons and propagating bounds over $f$ using the construction employed by Ladner et al. This neuron-merging construction fits this use case since it computes an outer approximation of sets over neurons in layer $L_i$ based on layer $L_{i-1}$. We can then prove that by propagating these approximations recursively through the neural network, the (newly defined) sufficiency condition for $f’$ is strictly stronger than the original sufficiency condition of $f$. While we acknowledge the reviewer's point that this proof is not very technically complicated, we highlight the subtlety in recognizing that the careful formulation of this construction for an abstract explanation in the context of the abstract model $f’$ allows us to establish this novel relationship between the sufficiency conditions of $f$ and $f’$, which allow certifying the sufficiency of $f’$ rather than $f$ (much more efficiently).
> >
> >
> >
> >
> > That being said, we agree with the reviewer’s initial observation that the greater challenge in obtaining the provable explanations discussed in this work lies not only in ensuring sufficiency but also in ensuring both sufficiency and *minimality* guarantees. Addressing this requires incorporating our novel “explanation refinement” mechanism that relaxes abstraction constraints at varying levels, thereby enabling provable minimality, as demonstrated in Algorithm 2. This process indeed forms the main bulk of this work and allows us to generate significantly concise sufficient subsets even at notably coarse abstraction levels.
> >
> >
> > We thank the reviewer again for highlighting this point, and we hope our answer has made this point clearer.

---

> > > ### Comment · Reviewer_shdo · 2024-11-25
> > > **Final reviewer comment**
> > >
> > > Thanks for the clarification.
> > >
> > > Overall, I'm happy with the discussion with the authors, as they addressed all my concerns and showed that the discussion was indeed useful in clarifying some points that were not clear before. I also checked other reviewers' comments, and I found myself in line with their opinions. I also double-checked the paper and as a final comment, I kindly ask the authors to clarify a bit more the paragraph Abstract Neural Networks in lines 210+, as I found it difficult to intuitively visualize the domain of the abstract network, which is defined over intervals instead of the real plane.
> > >
> > >
> > > I lean toward increasing my score to 8 as a sign of appreciation. However, I encourage the AC to take my recommendation cum grano salis, as I pointed out that formal verification is not my area of expertise.
> > >
> > >
> > > That said, best of luck with your submission!

---

> > > > ### Author Response · Authors · 2024-11-25
> > > >
> > > > We thank the reviewer for raising their score to an 8 and for encouraging us to improve the clarity of our paper further.
> > > >
> > > > We fully agree with the reviewer’s suggestion that incorporating illustrations to clarify the process of obtaining explanations for abstract networks will enhance the clarity of some concepts discussed in our paper. We will include these visualizations in the final version.
> > > >
> > > > Thank you once again for the valuable feedback!

---

### Official Review · Reviewer_ovvZ · 2024-11-02

**Soundness:** 3
**Presentation:** 3
**Contribution:** 2
**Rating:** 5
**Confidence:** 3

**Summary:**

This paper introduces an abstraction-refinement approach to efficiently generate provably sufficient explanations for neural network predictions. Traditional formal explainability methods are computationally intensive and struggle with scalability on larger models. This method simplifies the neural network by creating a smaller, abstract version, which speeds up the computation of explanations. If the explanation is insufficient, the network is gradually refined until a minimal, sufficient subset of features is identified.

**Strengths:**

1. The paper presents a unique abstraction-refinement technique, effectively bridging the gap between interpretability and scalability in neural network explanations. This approach is innovative in the realm of provable explanations.
2. Unlike many heuristic-based explainability methods, this technique provides formal guarantees for explanation sufficiency, which is highly valuable in safety-critical applications requiring reliability in interpretability.

**Weaknesses:**

1. The paper primarily demonstrates results on relatively simple models and standard datasets. Testing on more complex architectures (e.g., deep CNNs or Transformer-based models) would strengthen the claims regarding scalability and broader applicability.
2. While the abstraction-refinement approach reduces computational load, it is still constrained by the scalability limits of neural network verification. The authors could address this limitation by discussing ongoing advancements in verification techniques and how they might enhance this method.
3. The comparison focuses primarily on heuristic-based methods (e.g., Anchors and SIS) but lacks depth regarding alternative formal explanation methods. Adding comparisons with other provable techniques would provide a more comprehensive evaluation.
4. The abstraction process may lead to information loss, which could affect the explanation's precision or fidelity. The paper could benefit from a more in-depth analysis of the trade-offs between explanation minimality and information retention across abstraction levels.

**Questions:**

1. How well does the abstraction-refinement approach scale to more complex architectures, such as Transformers or deeper CNNs, beyond the datasets tested? Can the authors provide insights or preliminary results on its performance with larger models?
2. The paper mentions that abstraction reduces the network size, potentially losing information. How does this information loss impact the quality or trustworthiness of the explanations? Could the authors quantify or analyze this trade-off?
3. The experiments focus on image datasets. How does the approach generalize to other types of data, such as time-series, tabular data, or text? Are any modifications to the method necessary for non-image data?

---

> ### Author Response · Authors · 2024-11-18
>
> We thank the reviewer for the insightful comments. Our responses are provided below.
>
> **Extension to other non-image model types such as language models, and tabular data**
>
>
> Our method is data-type agnostic and can be applied to other model types, such as those handling text or tabular data. We chose to focus on vision tasks due to their suitability for visualizing results and the computational challenges they present, given the relatively large input space. A key consideration in language tasks is that an $\epsilon$ perturbation may significantly alter the semantic meaning of the input, making it less meaningful compared to vision tasks. An alternative approach is to apply $\epsilon$ perturbations within a latent space.
>
>
> We conducted an initial experiment using our method on the SafeNLP language task from the annual Neural Network Verification Competition [5]. This task, the only language-focused challenge in the competition, is based on the medical safety NLP dataset. Certification with respect to $\epsilon$ was achieved over an embedded representation of the input space, allowing for meaning-preserving perturbations. The experiment produced provably minimal and sufficient explanations, achieving a computation time of 0.66 ± 0.22 seconds and an explanation size of 6.67 ± 5.06. We appreciate the reviewer's suggestion to explore this direction and will include a comprehensive experiment in the final version of our paper to demonstrate our method's applicability to this benchmark.
>
>
> Another possible extension of our method is to regression tasks and not just classification. The provable guarantee here would be to satisfy that fixing the subset $S$ determines that the prediction remains within some $\delta$ range. Following a suggestion by reviewer eaKy, we demonstrated this on the real-world, safety-critical Taxi-Net benchmark employed by the work of Wu et al. [5]. The results demonstrate that our method produces explanations for this setting with an average computation time of 35.71 ± 3.71 seconds and an average explanation size of 699.30 ± 169.34, marking a significant improvement over the outcomes reported by Wu et al.
>
>
>
>
> We appreciate the reviewer for bringing up this point and will incorporate both of these experiments, demonstrating potential extensions of our method, into the final version.
>
> **Scaling the framework to larger architectures**
>
>
> Obtaining provable sufficient explanations indeed depends on neural network verification queries, which currently constrain the scalability of this approach for SOTA architectures. However, with rapid advancements in verification techniques [1 - 4, and as explained below], ongoing improvements in scalability will directly enhance the applicability of our method. Compared to other methods addressing the same task of providing provably sufficient explanations for neural network predictions [5 - 7], our abstraction-refinement approach is significantly more scalable. This is demonstrated not only by the substantial efficiency improvements highlighted in the experiments section but also by the generation of explanations over much larger models. For instance, our approach handles notably larger models compared to recent work providing provably sufficient explanations (Wu et al., NeurIPS 2023) [5], which follows a "traditional" methodology (i.e., it does not leverage our abstraction-refinement technique).
>
>
>
>
>
>
>
>
>
>
>
>
>
>
>
>
>
>
>
>
>
>
>
>
> **Providing background on neural network verification**
>
>
> Neural network verification is an active research area (e.g., [1–4]), with recent advancements in scaling methods like branch-and-bound [1,2,4] and abstract interpretation [3,4]. Our neuron-merging abstraction method is verification-agnostic, enhancing scalability in a complementary way.  In the final version, we will include a detailed overview of recent advancements to improve accessibility. Thank you for highlighting this.

---

> > ### Author Response · Authors · 2024-11-18
> >
> > **Comparison to other formally provable types of explanations, other than sufficient explanations**
> >
> >
> > While we compare our approach to heuristic and formally provable methods for computing sufficient explanations, directly comparing it to other provable explanation types - like minimal contrastive explanations or exact Shapley values - is inherently unfair due to differing objectives. A fair comparison would require methods identifying a sufficient subset of input features, which our experiments already address comprehensively.
> >
> >
> > However, in response to reviewer comments, including ovvZ and t2CN, we will add experiments to evaluate our approach across additional comparable configurations: brute-force occlusion, varied feature orderings (descending importance in additive attributions), and feature selection within an additive framework. Initial results comparing our approach to brute-force occlusion showed that the average subset sufficiency of that approach is 19%, similar to heuristic methods such as Anchors and SIS, with an explanation size of 14.92 ± 8.64. In contrast, our method inherently guarantees 100% sufficiency. Thank you for highlighting this.
> >
> >
> >
> >
> >
> >
> >
> >
> >
> >
> >
> >
> > **Information loss in the different abstraction levels may affect the explanation’s precision**
> >
> >
> > Our method produces significantly smaller networks, but the abstraction ensures that explanations that are provably sufficient for the reduced network *remain provably sufficient* for the original model, hence preserving the explanation’s precision.
> >
> >
> > However, it is true that a minimal sufficient explanation from the abstract model, while provably sufficient for the original model, may not remain provably minimal for it. This underscores the need for refinement. In a way, the explanation's size can indicate potential information loss at that level of abstraction, with larger explanations suggesting greater loss. However, we agree that traditional metrics like accuracy drop and KL divergence are valuable for measuring information loss. Comparing these to changes in explanation size offers an interesting perspective, which we will explore in an experiment in the final version.
> >
> >
> >
> >
> >
> >
> >
> >
> > [1] Beta-crown: Efficient bound propagation with per-neuron split constraints for neural network robustness verification (Wang et al., Neurips 2021)
> >
> > [2] Complete Verification via Multi-Neuron Relaxation Guided Branch-and-Bound (Ferrari et al., ICLR 2022)
> >
> >
> > [3] Differentiable abstract interpretation for provably robust neural networks (Mirman et al., ICML 2018)
> >
> >
> >
> >
> > [4] First three years of the international verification of neural networks competition (VNN-COMP) (brix et al., STTT 2023)
> >
> >
> > [5] Verix: Towards verified explainability of deep neural networks (Wu et al, Neurips 2023)
> >
> >
> >
> > [6] Distance-Restricted Explanations: Theoretical Underpinnings & Efficient Implementation (Huang et al., KR 2024)
> >
> >
> > [7] Towards Formal XAI: Formally Approximate Minimal Explanations of Neural Networks (Bassan et al., TACAS 2023)

---

> > > ### Comment · Reviewer_ovvZ · 2024-11-26
> > >
> > > Thank you for responding to the questions. Most concerns are resolved properly. I am willing to consider rasing my score.

---

> > > > ### Author Response · Authors · 2024-12-02
> > > >
> > > > Dear Reviewer ovvZ,
> > > >
> > > >
> > > > Thank you once again for your detailed and insightful feedback, which has been instrumental in identifying areas where our paper could be further clarified.
> > > >
> > > >
> > > > We are glad to have addressed your concerns and appreciate that you are considering increasing your score. Should you have any additional questions, we would be more than happy to address them.
> > > >
> > > >
> > > > Best regards,
> > > >
> > > >
> > > > The Authors

---

### Official Review · Reviewer_eaKy · 2024-11-02

**Soundness:** 4
**Presentation:** 3
**Contribution:** 3
**Rating:** 8
**Confidence:** 3

**Summary:**

The paper proposes an abstraction-refinement technique to create provably sufficient and minimal explanations for neural network predictions. The method works by initially generating a smaller, abstract network through a neuron-merging technique. Neurons with similar behavior are combined, controlled by a reduction rate, which specifies the extent of abstraction. This smaller network allows faster computation of explanations, but if the explanation is insufficient, the network is iteratively refined by gradually adding neurons back until a satisfactory explanation is found.

The evaluation on datasets like MNIST, CIFAR-10, and GTSRB shows that this approach is more efficient in both computation time and explanation size than traditional verification-based methods. However, the method’s reliance on neural network verification may limit scalability, and its testing on only standard benchmarks raises questions about real-world applicability. Nonetheless, the paper’s contribution lies in using formal verification to ensure that the explanations are very sound and reliable, which is critical for safety-sensitive domains.

**Strengths:**

1. Introduces a novel abstraction-refinement framework integrated with neural network verification for generating provable explanations, a unique combination in the field.
2. Empirical results are robust, with detailed comparisons to baseline methods.
3. The paper is well-written with precise definitions and a logical flow that enhances its readability and understanding.
4. Addresses a critical challenge in XAI, making it possible to deploy more reliable AI systems in regulated environments.

**Weaknesses:**

1. The method’s reliance on neural network verification queries limits its scalability to real-world applications. The verification tools required here may not support the level of scalability needed for larger, state-of-the-art networks.
2. The paper primarily tests on MNIST, CIFAR-10, and GTSRB, standard benchmarks that do not adequately test the scalability and generalizability of the method. This narrow evaluation undermines claims of efficiency and limits insights into practical, diverse applications (such as regression problem). Including a challenging regression problem, such as the aircraft taxiing task evaluated by Wu et al., would provide stronger evidence of the method's scalability and applicability in high-stakes, continuous-output domains.
3. The abstraction process risks oversimplifying the neural network to a degree that explanations may lose meaningful detail, leading to explanations that are formally sufficient but practically uninformative.
4.The paper’s current evaluation lacks a comprehensive set of baselines, particularly from perturbation-based and gradient-based explainability methods. Including comparisons with these widely used XAI techniques would better contextualize the capabilities of the proposed abstraction-refinement approach.

**Questions:**

1. Could the authors provide further insights or potential solutions on how to extend the applicability of their method to more complex, state-of-the-art neural network architectures?
2. In the experimental setup, was the computational overhead of the abstraction-refinement process compared to traditional methods quantified beyond explanation size and time? A breakdown of this could enhance the paper's impact.
3. How does the method perform across different domains, such as vision versus text? Are there domain-specific challenges that might affect the sufficiency of explanations?
4. Have the authors considered evaluating their method on a complex regression problem, such as the aircraft taxiing task used in Wu et al.'s work?

---

> ### Author Response · Authors · 2024-11-18
>
> We appreciate the reviewer’s valuable comments. See our responses below.
>
> **Reliance on neural network verification and extension to additional benchmarks and domains, such as Taxinet**
>
>
> We agree and acknowledge that any approach seeking to deliver provably sufficient explanations is naturally limited by the scalability challenges of neural network verifiers [1,2,3]. Nonetheless, progress in this field is advancing rapidly [4,5], and as the reviewer noted, our abstraction-refinement methodology provides a substantial improvement in the capability of using such methods to obtain provable explanations for neural networks. Additionally, we note that the benchmarks used in our experiments are more than one order of magnitude larger than those utilized in the notable work by Wu et al. (NeurIPS 2023, [1]), highlighting the improvements of our method in this regard as well.
>
>
> Additionally, similar to Wu et al., our framework can be adapted to regression tasks by verifying whether fixing a subset $S$ keeps the prediction within a specified $\delta$ range. Based on the reviewer's suggestions, we conducted an experiment using our approach on the TaxiNet benchmark. The results show that our method generates explanations for this setting with an average computation time of 35.71 ± 3.71 seconds and an average explanation size of 699.30 ± 169.34, representing a significant improvement compared to the results reported by Wu et al. We appreciate the reviewer’s suggestion to extend our results for this benchmark and will include a detailed analysis in the final version.
>
>
>
>
> Following another suggestion from the reviewer, we evaluated our method on the SafeNLP language task from the annual Neural Network Verification Competition [5]. This task, the only language task in the contest, is trained on the medical safety NLP dataset. The $\epsilon$ certification is achieved over an embedding of the input space, enabling meaning-preserving perturbations. Our experiments yielded the following results: an average computation time of 0.66 ± 0.22 seconds and an average explanation size of 6.67 ± 5.06. We appreciate the reviewers' suggestions and will incorporate an extended experiment on this benchmark in the final version to demonstrate the applicability of our method across additional domains.
>
>
>
>
>
>
> **Abstraction risks oversimplifying models to a degree where explanations are uninformative**
>
>
> We agree that this might partially occur at very low reduction rates, where the explanations remain provably sufficient at each abstraction level for the original model but are not necessarily provably minimal. However, our evaluations demonstrate that even at slightly higher reduction rates - still considerably smaller than the original network (such as around 20%-30% of the non-linear activations) - the provided explanations closely match those generated for the original model. This highlights minimal information loss in this context even at moderately higher reduction rates. This characteristic is, in fact, the core insight enabling our method to greatly enhance explanation generation efficiency, as most features are validated within coarser abstractions.
>
>
> **Comparison to additional XAI benchmarks**
>
>
> To ensure a fair comparison, our generated explanations should be evaluated against methods that also aim to identify subsets intended to be sufficient. This stands in contrast to most existing XAI techniques, such as classic additive feature attribution methods or many gradient-based methods, which do not focus on obtaining sufficient subsets and are therefore not directly comparable. Within this context, we believe that the comparisons in our work are thorough, as they include evaluations against the two most prominent heuristic approaches as well as traditional provable methods.
>
>
>
>
>
>
> However, in response to the reviewer’s comment, as well as the suggestions from reviewers t2CN and ovvZ, we will incorporate additional experiments to evaluate our approach across additional configurations. These experiments will include brute-force occlusion, provable explanations with varying feature orderings (based on descending importance in different additive attributions), and feature selection within an additive feature attribution framework. Preliminary results comparing our approach to brute-force occlusion revealed that the average subset sufficiency of the latter is 19%, comparable to heuristic methods like Anchors and SIS, with an explanation size of 14.92 ± 8.64. In contrast, our method naturally ensures 100% sufficiency.

---

> > ### Author Response · Authors · 2024-11-18
> >
> > **Did you compare abstraction-refinement to traditional methods using metrics other than time and size?**
> >
> >
> > Our primary evaluation method builds on prior works addressing sufficiency-based explanations [1,2,3], focusing primarily on the most common metrics: computation time and explanation size. Additionally, we present a detailed analysis that evaluates our results across different levels of abstraction, highlighting how computation time and explanation size are influenced by the level of abstraction.
> >
> >
> > Furthermore, in response to reviewer feedback and suggestions, we will include in our final review: (1) an analysis of how varying $\epsilon$ impacts the sufficiency of the produced subset, providing a more fine-grained understanding of both our method and heuristic approaches, along with their efficiency and explanation size, (2) an examination of how different feature orderings influence the resulting explanation, and (3) an evaluation of information loss across various abstraction levels.
> >
> >
> > We ran a preliminary experiment over MNIST with small $\epsilon$ perturbations to demonstrate their impact on computation time and explanation size:
> >
> >
> >
> >
> > | Perturbation Radius | Explanation Size | Computation Time |
> > |---------------------|------------------|-------------------|
> > |               0.012 | 219.450 ± 142.228 |  110.111 ± 33.712 |
> > |               0.011 | 186.970 ± 140.435 |  101.881 ± 41.625 |
> > |               0.010 | 153.240 ± 135.733 |  94.897 ± 46.244  |
> > |               0.009 | 119.040 ± 127.271 |  81.889 ± 52.578  |
> > |               0.008 | 87.530 ± 113.824 |  62.607 ± 58.084  |
> > |               0.007 |  59.420 ± 95.607 |  53.072 ± 56.709  |
> >
> >
> > We thank the reviewer for bringing up this point and will incorporate the full experiment on additional benchmarks, along with the other mentioned evaluations, in the final version.
> >
> >
> >  [1] Verix: Towards verified explainability of deep neural networks (Wu et al, Neurips 2023)
> >
> > [2] Distance-Restricted Explanations: Theoretical Underpinnings & Efficient Implementation (Huang et al., KR 2024)
> >
> >
> > [3] Towards Formal XAI: Formally Approximate Minimal Explanations of Neural Networks (Bassan et al., TACAS 2023)
> >
> > [4] Beta-crown: Efficient bound propagation with per-neuron split constraints for neural network robustness verification (Wang et al., Neurips 2021)
> >
> >
> > [5] First three years of the international verification of neural networks competition (VNN-COMP) (brix et al., STTT 2023)

---

> ### Comment · Reviewer_eaKy · 2024-11-24
>
> Dear Authors,
>
> Thank you for your thoughtful response. Similar to my works, I find this research demonstrates significant potential in utilizing formal methods for explainable AI. And I found it inspiring. I thoroughly enjoyed reading your paper and appreciate the effort you have put into expanding the dataset beyond my earlier suggestions, as my biggest concerns regarding scalability to other datasets and models are somewhat addressed.
>
> I do not have any further questions and would like to maintain my score as it is if the paper revision is reflected accordingly.

---

> > ### Author Response · Authors · 2024-11-25
> >
> > We sincerely thank the reviewer for their strong support of our paper and for acknowledging its significant contribution to advancing the possibility of obtaining explanations with formal guarantees for neural networks.
> >
> > We would like to highlight that we have included additional details about the mentioned experiments in the revised manuscript, along with other results requested by the reviewers (see general comment for more information).
> >
> > Once again, we deeply appreciate your valuable feedback!

---

### Official Review · Reviewer_t2CN · 2024-11-04

**Soundness:** 3
**Presentation:** 3
**Contribution:** 3
**Rating:** 8
**Confidence:** 3

**Summary:**

The work introduces an approach to provide "provably minimally sufficient
explanations", based on "abstraction refinement". Provably minimally
sufficient explanations are defined as the minimal set of features required to
under which the model produces the same prediction. Abstraction
refinement is a method from model checking and verification, which reduces the
complexity of the model while keeping the prediction (winning class) constant,
somewhat similar to model pruning. The work includes a formal proof motivating
its approach, and some empirical experiments analyzing the runtime and
identified number of minimal features, as well as a comparison to two similar
approaches with respect to sufficiency and runtime.

**Strengths:**

- The work is clearly written.

- The proposed approach is well-motivated.

- The work makes the limitations of its approach clear.

- The application of abstraction-refinement from the area of model verification
  to feature explanation in an occlusion-based context is original.

**Weaknesses:**

- Significance: The practicality of the approach is severely limited by is long
  runtime (up to 3 hours for a single attribution mask). This issue could be
  alleviated by discussing trade-offs, especially considering the approaches in
  Table 2. Comparing with these methods, SIS and Anchors, there is likely some
  optimal trade-off between sufficiency and runtime that would be valuable to
  analyze.

- Contribution: The identified explanations may not necessarily be unique. A purely additive
  model with some threshold could have any combination of inputs, as long as
  the threshold is passed (i.e., the prediction does not change). Due to this,
  the identified feature attributions might not necessarily present all
  features relevant for the prediction, but rather only a subset thereof.
  A discussion of the uniqueness, and the issue of the order of removing the
  features, would be very valuable.

- Novelty: There is a plethora of approaches (see, e.g., Ancona et al., 2019 for
  approximations of Shapley values) that assign relevance to features (somewhat
  different to choosing feature subsets) with this issue without constraining
  the sufficiency (i.e., the fidelity) of the model directly. These mostly
  avoid computing the Occlusion (see Zeiler 2014), which observes the
  prediction under removal of individual features, due to its infeasible
  runtime. The approach presented is very similar to occlusion-based
  approaches, as the model is reduced in order to occlude parts of the input.
  This is an important body of related work to discuss.


References:

Ancona, M., Oztireli, C., & Gross, M. (2019, May). Explaining deep neural
networks with a polynomial time algorithm for shapley value approximation. In
International Conference on Machine Learning (pp. 272-281). PMLR.

Zeiler, M. D. (2014). Visualizing and Understanding Convolutional Networks. In European conference on computer vision/arXiv (Vol. 1311).

**Questions:**

- Did you consider some trade-off of sufficiency versus runtime?

- How do you solve the issue of uniqueness of the relevant feature set?

- How does this work compare to "brute-force" occlusion?

- How did you verify the sufficiency for the heuristics-based approaches?

---

> ### Author Response · Authors · 2024-11-18
>
> We appreciate the reviewer’s valuable and thoughtful comments. Please find our responses below.
>
>
> **The practicality of the approach and exploring potential trade-offs between sufficiency and runtime.**
>
>
> Certifying the provable sufficiency of an explanation depends on the use of a neural network verifier, which currently poses scalability challenges. However, advancements in this area are rapid [1,2], and our approach significantly outperforms competing methods for the same task [e.g., 3 - Wu et al., Neurips 2023], including substantial improvements in runtime (as highlighted in the experiments section) and in handling significantly larger models. While methods such as SIS and Anchors are generally more scalable since they do not rely on verifiers, they cannot guarantee provable sufficiency. As shown in the experiments section, fewer than 20% of the subsets generated by these methods were sufficient, whereas our method consistently produces provably sufficient subsets by design.
>
>
>
>
> However, we do agree with the reviewer that discussing certain trade-offs is worthwhile, and we will include a thorough discussion of this in our final draft. Here is a brief overview:
>
>
> **Sufficiency-runtime tradeoff:** A potential sufficiency-scalability trade-off lies in the size of the $\epsilon$ ball for explanation verification. Smaller $\epsilon$ balls improve scalability but limit sufficiency verification to narrower domains. We performed an initial MNIST experiment with small $\epsilon$ perturbations, which showed how this reduction impacts computation time.
>
>
>
>
> | Perturbation Radius | Explanation Size | Verification Time |
> |---------------------|------------------|-------------------|
> |               0.012 | 219.450 ± 142.228 |  110.111 ± 33.712 |
> |               0.011 | 186.970 ± 140.435 |  101.881 ± 41.625 |
> |               0.010 | 153.240 ± 135.733 |  94.897 ± 46.244  |
> |               0.009 | 119.040 ± 127.271 |  81.889 ± 52.578  |
> |               0.008 | 87.530 ± 113.824 |  62.607 ± 58.084  |
> |               0.007 |  59.420 ± 95.607 |  53.072 ± 56.709  |
>
>
>
>
> In the final draft, we will provide a thorough analysis of this tradeoff, determined by the $\epsilon$ perturbation, across all our benchmarks.
>
>
>
>
> **Minimality-runtime tradeoff:** Another possible trade-off concerns minimality. Our approach shows that provably sufficient explanations can be achieved at varying levels of abstraction. Even with low reduction rates (e.g., reducing 90% of non-linear activations), many input dimensions can be verified, yielding a concise subset. Users can halt the abstraction-refinement process early, generating a provably sufficient (though non-minimal) explanation with significantly higher efficiency - requiring only ~10% of the computation time compared to the full algorithm. We thank the reviewer for raising this and will highlight it in the final version.
>
>
> **Methods for abstraction in additive attributions and occlusion-based approaches should be referenced**
>
>
> We acknowledge the reviewer's emphasis on the importance of addressing methods, such as the notable work by Ancona et al., which employ probabilistic or uncertainty-based abstractions of neural networks to constrain additive attribution fidelity. However, we note that while these works focus on *probabilistic* guarantees, our method provides a much stronger, strict formal guarantee over entire continuous domains. To our knowledge, this is the first abstraction algorithm over neural networks to offer explanations with such guarantees. Our method introduces additional novelty through the refinement component, which progressively relaxes the abstraction constraints, and hence also ensures provable minimality guarantees.
>
>
>
>
> Similarly, we acknowledge that occlusion-based methods, such as those by Zeiler et al., are also relevant for their focus on feature subsets, and we will discuss them as related work. However, we do note a fundamental difference between occlusion-based methods and the formal guarantees provided by our approach. Specifically, occlusion methods typically rely on a fixed auxiliary baseline for the occluded complement, whereas our method ensures the strict sufficiency of the derived subset across an entire continuous domain, offering significantly stronger formal guarantees.

---

> > ### Author Response · Authors · 2024-11-18
> >
> > **Further discussion is needed regarding the (non)-unique generated subsets and the feature orderings of our method**
> >
> >
> > We thank the reviewer for highlighting the importance of feature ordering and the non-uniqueness of generated subsets, a common trait in methods for this task. We do not discuss this thoroughly since it is not the primary focus of our work and has been explored in prior studies [3,4,5]. However, we agree with the reviewer on its importance and will address it thoroughly in the final draft:
> >
> >
> > We first highlight that while sufficient explanations are indeed not unique, their intrinsic characteristics enable them to capture aspects often missed by additive attribution methods, such as feature interactions and non-linear behaviors. For example, the authors of Anchors [6] demonstrate that this type of explanation frequently yields results that are more intuitive and preferred by humans.
> >
> >
> >
> >
> >
> >
> > The widely adopted approach [3,4,5] (which we also use) for obtaining subsets that are both *concise* and where the uniqueness concern is less dominant involves sorting features in descending order of their attributions and progressively removing the least significant ones first, as they are less likely to impact the classification outcome. This has the advantage of converging towards *smaller* subsets as well as to subsets that substantially *overlap* with other sufficient explanations, and hence mitigate the possible discrepancy between a random ordering over features (the “uniqueness” concern).
> >
> >
> > In this work, we follow the ordering proposed in [3] that prioritizes features based on their (descending) sensitivity values. Following the reviewer’s comments, we will include an experiment that demonstrates the results of our approach under various feature orderings, leading to convergence toward different minimal subsets and studying the overlap of these subsets. We thank the reviewer for raising these points.
> >
> >
> >
> >
> >
> >
> >
> >
> >
> >
> >
> >
> > **Comparison to brute-force occlusion**
> >
> >
> > As mentioned earlier, our approach provides significantly stronger formal guarantees than traditional occlusion-based methods that rely on auxiliary baselines. To illustrate, we performed a brute-force occlusion experiment on MNIST with a fixed baseline of $0$, using the same pixel occlusion order as our algorithm. The average subset sufficiency was 19%, similar to heuristic methods such as Anchors and SIS, with an explanation size of 14.92 ± 8.64. In contrast, our method inherently guarantees 100% sufficiency. We will include this experiment for all benchmarks in the final draft. Thank you for the suggestion.
> >
> >
> >
> >
> >
> >
> >
> >
> > **How did you verify the sufficiency of the heuristic-based approaches?**
> >
> >
> > Since the heuristic-based approaches examined generate a subset of input features as their final output, we can evaluate the sufficiency of these subsets using the same procedure applied to verify our generated subsets. This involves validating their capacity to maintain sufficiency within an $\epsilon$ perturbation, which can be integrated with a neural network verifier.
> >
> >
> > [1] Beta-crown: Efficient bound propagation with per-neuron split constraints for neural network robustness verification (Wang et al., Neurips 2021)
> >
> >
> > [2] First three years of the international verification of neural networks competition (VNN-COMP) (brix et al., STTT 2023)
> >
> >
> > [3] Verix: Towards verified explainability of deep neural networks (Wu et al, Neurips 2023)
> >
> >
> > [4] What made you do this? Understanding black-box decisions with sufficient input subsets (Carter et al., AI’STATS 2019)
> >
> >
> > [5] Distance-Restricted Explanations: Theoretical Underpinnings & Efficient Implementation (Huang et al., KR 2024)
> >
> >
> > [6] Anchors: High-precision model-agnostic explanations (Ribeiro et al., AAAI 2018)

---

> > > ### Comment · Reviewer_t2CN · 2024-11-26
> > >
> > > Thank you for your extensive and thoughtful response.
> > >
> > > I highly appreciate the discussion with respect to the trade-off, and the added
> > > the experiment with the decreasing $\epsilon$ perturbation radius. I am happy
> > > to read that the approach allows to trade runtime for explanation quality.
> > > I also acknowledge the fundamental difference to the works by Ancona et al. and
> > > Zeiler et al. differ in the formal guarantees presented in this manuscript,
> > > which nonetheless I appreciate the authors to discuss. Further, I am looking forward
> > > to see the full results for brute-force occlusion. Although the low sufficiency
> > > was to be expected, I appreciate the addition as a fairly intuitive baseline.
> > >
> > > By also including the other reviews and respective replies, I feel confident in
> > > raising my score to 8.

---

> > > > ### Author Response · Authors · 2024-11-27
> > > >
> > > > We thank the reviewer for increasing their score to an 8, as well as for the insightful feedback and suggestions.
> > > >
> > > > We will certainly incorporate the recommended aspects into our final version, including a comprehensive comparison with the occlusion-based benchmark.
> > > >
> > > > Once again, we thank the reviewer for their valuable input!

---

### Author Response · Authors · 2024-11-25

Dear Reviewers,

Thank you once again for your insightful feedback and for recognizing the significance of our work. We have addressed specific additional experimental and theoretical results in individual discussion threads. For your convenience, we have also incorporated these updates into the revised manuscript, including:

1. We conducted an additional experiment to evaluate our method on the safeNLP benchmark from the annual neural network verification competition [1] and the real-world regression Taxinet benchmark, previously studied by Wu et al. ([2], NeurIPS 2023). These experiments were carried out to demonstrate the applicability of our method to additional domains, following the suggestions from reviewers eaKy and ovvZ (attached in Appendix D).

2. Further ablation studies on varying $\epsilon$ perturbations to better illustrate sufficiency-scalability trade-offs, as suggested by Reviewer t2CN (attached in Appendix D).

3. Additional ablations of our results based on varying feature orderings, as recommended by Reviewer t2CN (attached in Appendix D).

4. A complete and more rigorous refinement of the proof for Corollary 1, as suggested by Reviewer dQgK (attached in Appendix A).

Additional detailed responses are provided in the respective threads. We sincerely appreciate your valuable feedback and are happy to discuss any further points.

[1] First three years of the international verification of neural networks competition (VNN-COMP) (brix et al., STTT 2023)

[2] Verix: Towards verified explainability of deep neural networks (Wu et al, Neurips 2023)

---

### Meta-Review · Area_Chair_oM6c · 2024-12-23

**Metareview:**

This paper proposes an abstract refinement framework based on neural network verification to obtain provably sufficient explanations of neural network predictions with improved efficiency. The key idea is to leverage the neuron-merging-based abstraction to obtain sufficient explanation for abstract network and derive the abstract sufficient explanation, which will be sufficient for the original network. However, the main/core result, abstract sufficient explanation, in sec 3 is immediate if one using the abstract interpretation (e.g. diff AI), linear bounding framework (e.g. crown) results in neural network robustness verification, and there are many follow-up work along this line, which can handle more complicated and larger scale models than the models studied in this work (e.g. in Appendix C). However, the current work missing this important part of literature for both experiment comparison and discussion, which is essential for the topic to use formal verification tool for post-hoc explanations.

**Additional Comments On Reviewer Discussion:**

In the rebuttal period, there are some major discussions regarding the run-time, scalability of the proposed method. The authors provided additional computation results over mnist with different perturbation size to show the sufficiency-runtime trade-off. The authors also acknowledge the scalability is a major challenge in the formal verification field, nevertheless, their results is better than the prior work (Wu, NeurIPS 23).

---

### Decision · Program_Chairs · 2025-01-22

Reject